



# Declining Sea Ice and Its Relationship with Arctic Cyclones in Current and Future Climate Part I: Current Climatology in CMIP6 Models

Elina Valkonen[1,2,3, 4], John Cassano[1,2,3], Elizabeth Cassano[1,2], and Mark Seefeldt[1,2]

[1]National Snow and Ice Data Center, University of Colorado Boulder, Boulder, Colorado, United States
[2]Cooperative Institute for Research in Environmental Sciences, University of Colorado Boulder, Boulder, Colorado, United States
[3]Department of Atmospheric and Oceanic Sciences, University of Colorado Boulder, Boulder, Colorado, United States
[4]Department of Atmospheric Science, Colorado State University, Fort Collins, Colorado, United States
**Correspondence:** Elina Valkonen (elina.valkonen@colorado.edu)

**Abstract.** The Arctic climate system is changing rapidly. These large changes will have implications in the Arctic and beyond. One of the main components of the Arctic climate system are Arctic cyclones. The strong coupling between the sea ice and Arctic cyclones makes it an important topic in the warming climate. In this study, an ensemble of CMIP6 model output was utilized from 1985 - 2014, to determine how well the chosen models depict Arctic cyclones and their relationship with sea ice.

A comprehensive climatology of Arctic cyclones and sea ice concentrations (SIC) was provided based on selected models from CMIP6 and the results were compared to the ERA5 product. The model results did closely match reanalysis data in depicting the observed sea ice trend. However, we found that the model results struggled to reproduce the strongly coupled relationship between the declining sea ice and Arctic cyclones. The local cyclogenesis in the Arctic was shown to be underestimated, which led to an overall underestimation of Arctic cyclones in the CMIP6 model results. The results also showed differences between

model results and ERA5 with regard to cyclone intensities. The the magnitude and sign of the intensity differences varied based on the nominal resolution of the model, their surface roughness parametrization and cyclogenesis location.

## 1   Introduction

Dramatic changes in the Arctic climate system have been observed in recent years. Since the beginning of the satellite record (1979) the 14 lowest sea ice extent years have occurred in the past 20 years, with the record low sea ice extent observed in

2012 (NSIDC, 2020). Surface air temperature records have been broken in the high North in recent years along with the rest of the globe (Ballinger at al., 2020). This warming in the Arctic and consequent rapid sea ice decline, a phenomenon known as Arctic Amplification (Ballinger et al., 2020, Maslanik et al., 2007; Stroeve et al., 2008; Screen Simmonds, 2010; Stroeve et al., 2012; Onarheim et al., 2018), has large consequences on the physical, biological and social components of the Arctic.

Changes in the Arctic climate and associated sea level rise can also cause storms to be more threatening to Arctic coastal

communities due to increasing storm surge (Lantz et al., 2020). Some studies have predicted that by 2050 the Arctic will be ice free in the summer (Onarheim et el., 2018), allowing for more human activities in the high North. The importance of accurate



weather and sea ice forecasts for the Arctic region for the safety of coastal communities, and the future economics and tourism cannot be overstated.

The declining sea ice has been the focus of many observational and modeling studies, such as Maslanik et al. (2007), Stroeve et al. (2008), Serreze et al. (2009), Bader et al. (2011), Serreze and Meier (2018) and Simmonds and Li (2021). All these studies have demonstrated the consistent steep decline of the Arctic sea ice using multiple observational products and models. The representation of Arctic sea ice decline and Arctic Amplification in models has also been studied. For example, Ye and Messori (2021) compared the surface air temperature (SAT) and SIC trends in an ensemble of 6th generation Coupled Multimodel Intercomparison Project (CMIP6) data to ERA-Interim and CESM-LE results. They found that the even though all the models did depict a decline in the SIC, the models show a large spread in SIC results, partly due to large internal climate variability and were less consistent with the ERA-Interim results than the SAT.

An important element in the Arctic climate system are extratropical cyclones (Serreze and Barry, 2014). Arctic cyclones either travel to the Arctic from midlatitudes or are generated locally in the Arctic (Sepp and Jaagus, 2011; Serreze and Barrett, 2008; Zhang et al., 2004). This leads to multiple different types of cyclones in the Arctic, from the synoptic scale midlatitude cyclones to mesoscale polar lows, and to the observed summer maximum in cyclone counts over the Arctic (Serreze and Barret, 2008; Crawford and Serreze, 2016; Valkonen et al., 2021). Cyclones play a major role in the climate system in general by transporting energy, moisture, and momentum from the Equator to the poles (Bader et al., 2011; Ulbrich et al., 2009; Varino et al., 2018) and defining large parts of the weather we experience in the mid-and high latitudes (Browning 2004; Hawcroft et alt., 2012). In the Arctic, cyclones influence the energy exchange between the ocean, sea ice and atmosphere and the atmospheric and oceanic boundary layers, highlighting their critical role in the coupled Arctic climate system (Bengtson et al., 2011, 2013; Zahn et al., 2018). In addition, the relationship between Arctic cyclones and sea ice is strongly linked (Simmonds and Keay, 2009; Valkonen et al., 2021). The complex relationship between cyclones and the changing sea ice, and cyclones' important role in the Arctic now and in the future, make it critical to better understand the interactions between Arctic cyclones and sea ice, and how these interactions may change with a warming climate.

Many previous studies have focused on Arctic cyclones in observational and reanalysis data, in the present and near recent history. For example, Zahn et al. (2018) used data from four different reanalysis products to construct a climatology of Arctic cyclone characteristics and studied their trends in the recent past. They found good agreement between the different reanalysis products and an increase in cyclone frequencies and depth with time over certain parts of the northern North Atlantic and central Arctic in the winter season. Similarly, a positive cyclone count trend was observed by Sepp and Jaagus (2011), who studied Arctic cyclone characteristics based on their entrance region to the Arctic.

Modeling studies have also been conducted to gain more insight into Arctic cyclones and their development. The current generation of climate models provide a more accurate representation of the Arctic climate system compared to the older climate models (Harvey et al., 2021; Priestly et al., 2020). Many previous studies have used ensembles of CMIP3 and CMIP5 simulations to study the accuracy of the participating models, the effect of recent warming in the Arctic and midlatitudes, as well as the general skill of the models, both in the Arctic and at midlatitudes (Harvey et al., 2021; Ye and Messori, 2021; Priestly et al., 2020; Song et al., 2021). Harvey et al. (2021) found that in general there are equatorward biases in the midlatitude storm





tracks in the CMIP3 and CMIP5 models leading to an extension of the North Atlantic storm track into Europe instead of turning poleward towards the Arctic. Similar results were found by Priestly et al. (2021), who used output from an ensemble of 20 CMIP6 models to study the accuracy of the CMIP6 simulations compared to the ERA5 data and the previous generation models

(CMIP5). The CMIP6 models have made some improvements in these areas, but underestimation of midlatitude cyclones in the NH remains, even though the bias has decreased in magnitude. Some of the modeling studies have also investigated Arctic cyclones in the CMIP5 and CMIP6 models. Song et al. (2021) found an underestimation of Arctic cyclone counts due to the zonal bias in the North Atlantic and Pacific storm tracks and due to issues with reproducing correct rates of Arctic cyclogenesis. Model resolution and coupling between the atmosphere and ocean in the models have been shown to play a major role in the

accuracy of representing Arctic cyclones (Small et al., 2019; Song et al., 2021; Zolina and Gulev, 2002).

Coupling between the atmosphere and ocean is especially important when assessing the relationship between sea ice concentration (SIC) and cyclones. These features are strongly intertwined as shown by Valkonen et al. (2021), which studied the representation of Arctic cyclones and the relationships between Arctic cyclones and sea ice in three different reanalysis products. They found that cyclone counts and certain intensity metrics were influenced by the SIC, but that the cyclones could

also influence the SIC depending on the season and the seasonal lags considered. Arctic sea ice decline and its relationship with storms has also been the focus of other observational and modeling studies (Deser et al, 2000; Overland and Pease, 1982; Shreiber and Serreze., 2020; Simmonds and Keay, 2009), with mostly similar results of increasing cyclone frequencies and intensities with declining sea ice, although some contradicting results were also found. For example, Schreiber et al. (2020) who studied the sea ice response to a passing cyclone using multiple reanalysis products found that SIC concentrations increased

after cyclone passages.

As just discussed, improvements have been made in modeling both sea ice and cyclones in the Arctic but further work is needed to understand the complex relationship between these two critical aspects of Arctic climate. Studies have also been conducted to better understand how this relationship might change with changes in Arctic climate. However, fewer studies have investigated this topic specifically in climate models by focusing solely on cyclones that exist over the Arctic Ocean and

how they interact with the sea ice. Also, only a few studies exist that have used the CMIP6 climate models to do this, which shows greater accuracy in representing Arctic and midlatitude cyclones than the previous CMIP data sets. In this study we use an ensemble of CMIP6 data to create a climatology of Arctic cyclones and study their relationship with Arctic SIC. The model results will be compared to each other and ERA5 reanalysis data to gain understanding of the ability of the models to recreate the Arctic cyclone characteristics, the Arctic SIC and their relationship. The results of this study will then be utilized

in a companion paper, which will investigate the Arctic cyclones and sea ice in the future SSP5-8.5 scenario. The main goals of this study are:

1. To describe the Arctic cyclone and SIC climatology based on an ensemble of CMIP6 models.

2. To quantitatively describe the biases in both the multi-model means and individual models in representing Arctic cyclone characteristics and SIC compared to a state-of-the-art reanalysis product (ERA5).





3. To assess the CMIP6 models' ability to represent observed relationships between Arctic cyclones and sea ice, and to accurately describe the causalities between the two.

The paper is constructed as follows: Section 2 describes the data and methods used in this study. Section 3 presents the results of our analysis and Section 4 provides a discussion of our findings and summarizes the main conclusions of this research.

## 2 Data and Methods

### 2.1 CMIP6 simulations

In this study CMIP6 historical simulations are used to assess how well the chosen models perform in representing Arctic cyclones, their characteristics and relationships between cyclones and Arctic sea ice. The CMIP6 historical runs span from the year 1850 to the present day and are forced with external forcing, such as solar variability and human induced changes in the greenhouse gas concentrations (Eyring et al., 2016). These forcings are based on observations and hence the historical simulations give a good representation of the models' performance. In this study six models were chosen from the CMIP6 project based on availability of the necessary data for the cyclone tracking and analysis. All the chosen models are fully coupled global models with interactive oceans and sea ice. The models came from multiple different modeling centers around the world. EC-Earth3 from Europe (Döscher et al., 2021), BCC-CSM2-MR from China (Wu et al., 2019), MRI-ESM2-0 (Yukimoto et al., 2019) and MIROC6 (Tatebe et al., 2019) from Japan, and MPI-ESM1-2-HR (Mauritsen et al., 2019; Müller et al., 2018) and MPI-ESM1-2-LR (Mauritsen et al., 2019) from Germany. These models were chosen because archived data at 6-hourly intervals was available for both the historical and SSP5-8.5 simulations. As the results of the analysis performed in this paper will be used as a basis for looking at future simulations (namely SSP5-8.5) data availability for that simulation was taken into account when choosing the subset of CMIP6 models to analyze for this work. Other CMIP6 models did not have high enough output frequency (for the SSP5-8.5 simulations) to be considered. For this analysis 6-hourly output was required for the sea level pressure (SLP), 10-meter zonal wind (U10M) and 10-meter meridional wind (V10M) to perform cyclone tracking and subsequent analysis. The sea ice concentration (SIC) was used at daily resolution, since sub-daily values were not available, and the SIC does not exhibit large daily variability. EC-Earth3, BCC-CSM2-MR, MRI-ESM2-0 and MPI-ESM1-2-HR all have 100 km native horizontal resolution. MIROC6 and MPI-ESM1-2-LR have lower resolution with 250 km native horizontal resolution. All the model data as well as the ERA5 data were regridded to a 50 km EASE grid . Results for the 30-year time period from 1985 to 2014 was used for the analysis presented here.

### 2.2 Reanalysis data – ERA5

In the Arctic region it is more difficult to gather observations than in the midlatitudes. Due to the harsh environment, it is hard to set up automated weather station networks or conduct observational campaigns. Satellite measurements help but have their own issues due to the unique conditions in the Arctic. As a result, observations in the Arctic are sparse. That is why reanalysis products, which combine state of the art models and available observations to produce gridded analyses of the recent past are





useful since they provide consistent gridded data across the Arctic and are more accurate than individual observations would be. In this study the ensemble of CMIP6 models were compared against the ERA5 reanalysis product (Hersbach et al., 2020), which is the newest reanalysis from the European Centre for Medium-Range Weather Forecasts (ECMWF) and is the highest global resolution reanalysis product to date. ERA5 has a horizontal resolution of 31 km (T639) and 137 vertical levels (up

to 0.01 hPa). The ERA5 product extends from 1950 to the present. In this study the time range used was the same as for the CMIP6 data, from 1985 to 2014. Valkonen et al. (2021) have shown that the ERA5 product performs well compared to the older reanalysis products.

## 2.3 Cyclone Tracking Algorithm

A cyclone tracking algorithm is applied to the ERA5 and the CMIP6 models' SLP fields. The tracking algorithm is described

in Crawford and Serreze (2016), which was updated from Serreze (1995) to track cyclones based on their minimum SLP. Each cyclone is identified by comparing the SLP at each grid point to its neighboring grid points. The identified cyclone is then followed through its lifecycle. This is done by giving each identified minimum a unique identifier and comparing identified SLP minima to the possible new locations of the existing cyclone. The possible new locations are calculated based on maximum propagation speed of 150 $kmhr^{-1}$, which is based on values from previous research (Wernli and Schwierz (2006)). If a match

is found the new location (longitude, latitude) is added to the track. As cyclones are not point objects this algorithm also returns the area of the cyclone, defined by the last closed isobar (2 hPa isobar interval is used) before another minimum or maximum is observed. This is a metric that is not output by most other tracking algorithms and is advantageous for the work done in this study. This allows us to calculate cyclone intensity and the average SIC over the cyclone area, which is more representative than looking at just a single point at the cyclone center. The tracking algorithm identifies and disregards unphysical cyclones

and cyclones that are too weak. For example, elevations above 1500 m are masked to avoid biased SLP values. Even so, a few non-realistic cyclones can be found in the tracking output. The biggest issue found was that there were cases were cyclone depth was found to be zero, together with an area size of one grid cell. These zero depth cases were excluded from our analysis. If a 'zero depth' case took place in the middle of the cyclone track, the track was split into two separate tracks.

Following Valkonen et al. (2021) the cyclone tracking results are post-processed to gain a better understanding of the Arctic

cyclones. The tracking algorithm runs globally, but the focus of this study is the Arctic region (shown in Figure 1), so only cyclones that exist for 24 hours or more in the study region were included in the cyclone matrix.

For these cyclones multiple characteristics, such as central pressure, lifetime, depth and cyclone area were recorder for every time-step over the cyclone's lifetime. In addition, SIC over the cyclone area and the cyclone central point were recorded. Based on these statistics each cyclone was classified based on the season they existed, cold (December-May) or warm (June-

November). These seasons were centered around SIC maximum and minimum months. How intense each cyclone was (weak, normal strong , calculated based on the 25th lowest, interquartile and the top 25th percentile values of ACE over the whole study period), and the average SIC (less than 15%, more than 85%, or in between) were also noted in the cyclone matrix.

To assess cyclone intensities multiple metrics were used. Cyclone central pressure, depth (difference between central pressure and pressure at edge of cyclone) and cyclone pressure gradient (DpDr) were returned by the tracking algorithm and gave



**Figure 1.** Map showing the study region with yellow outline (defined as "Arctic" in the text). Main locations mentioned in the study are also marked.





a good understanding of the cyclone strength. However, our focus is on the interactions between sea ice and cyclones and one critical aspect of this interaction is defined by the momentum and heat transferred between the surface and the cyclone. To capture this relationship more robustly we used the Accumulated Cyclone Energy (ACE) metric as an additional intensity metric for the cyclones. The ACE, as used in this study, is defined following Klotzbach (2006) but is calculated with the mean squared wind speeds over the cyclone area instead of the cyclone maximum wind speed. As shown in Valkonen et al. (2021),

the use of ACE based on the cyclone area wind speed or based on the maximum wind speed are very similar with the ACE based on cyclone area wind speeds giving more emphasis on the whole cyclone effect.

## 2.4 NAO-index

To understand how the models depict large-scale circulation patterns in the Arctic and if the differences in cyclone characteristics were forced by large-scale circulation or local processes, the North Atlantic Oscillation (NAO) index was used as a

proxy for the large-scale circulation pattern over the North Atlantic and Europe, where most cyclones enter the Arctic (Serreze and Barry, 2014). The NAO-index was calculated for each model and the ERA5 product following Hurrell and Deser (2009). This was done by calculating the eigenvectors, which were constrained to be spatially and temporally orthogonal to each other, of the cross-covariance matrix based on the seasonal SLP anomalies. These eigenvectors are then scaled based on the total variance they explain, and the principle component analysis (PCA) corresponding to the first eigenvector that explains the

most variability is the NAO-index. This method is known as empirical orthogonal functions or principal component analysis (EOF-PCA analysis) and is commonly used to define other climate indices. This analysis was performed over the North Atlantic (20°–70°N; 90°W–40°E). The advantages of using the EOF method over the traditional comparison of SLP between two weather stations (between the Icelandic low and Azores high) is that this method is not affected by the changes in the SLP baseline, and is therefore valid over all seasons, which the traditional method is not. Care must be used however when

interpreting the EOF analysis results as they return mathematical constructs that can have non-physical meaning. That is why the patterns of the first eigenvalue and the variance explained for ten first eigenvalues were plotted for all the models and ERA5 to make sure the results were physically reasonable; before the PCA time series (NAO index) was used for further analysis.

## 3 Results

### 3.1 Cyclone characteristics

Figure 2 depicts the seasonal variability in observed cyclone counts over the Arctic Ocean and the models' ability to reproduce the observed spatial patterns in the seasonal cyclone counts. Between 1.5 and 4.5 cyclones per 150 km x 150 km area are observed depending on the season and location. In ERA5 and the models, the cold season cyclones (panels 2a, c-h) are more concentrated on the North Atlantic side of the Arctic Ocean, whereas in the warm season (panels 2 b, i-n) most cyclones are located in the central Arctic. The difference in cyclone locations is related to a cyclone count maximum in the warm season,

which has been attributed to higher cyclogenesis rate in the Siberian plateau (Serreze et al., 2016) and is shown also to be





**Figure 2.** The seasonal average cyclone counts for 1985-2014 calculated over 150km x 150km grid boxes for cold and warm seasons.

related to a higher cyclogenesis rate in the Arctic warm season (as discussed later; Fig. 4b). Given the small number of grid points with significant differences in cyclone counts between the models and ERA5 there is little evidence for systematic differences in most models (Fig. A1). Although, all of the models do depict a slight tendency to underestimate cyclone counts in the central Arctic, which is evident by the scattered negative differences in each of the model plots. In the warm season the tendency of the models to underestimate the cyclone counts is more coherent and concentrated over the Central Arctic in several models. BCC-CSM2-MR, MIROC6 and MPI-ESM1-2-LR/HR are underestimating the cyclone counts by 20%-60% at selected grid points in the central Arctic in the warm season (panels A1, panels j, l and m-n), with BCC-CSM2-MR showing highest and most consistent underestimation of 60% (Fig. A1j).

There is a large interannual variability of Arctic cyclone counts present in all of the models as well as in the ERA5 product. While the 30 year mean basin wide cyclone count in the cold season in ERA5 is well reproduced by the models there is a lack





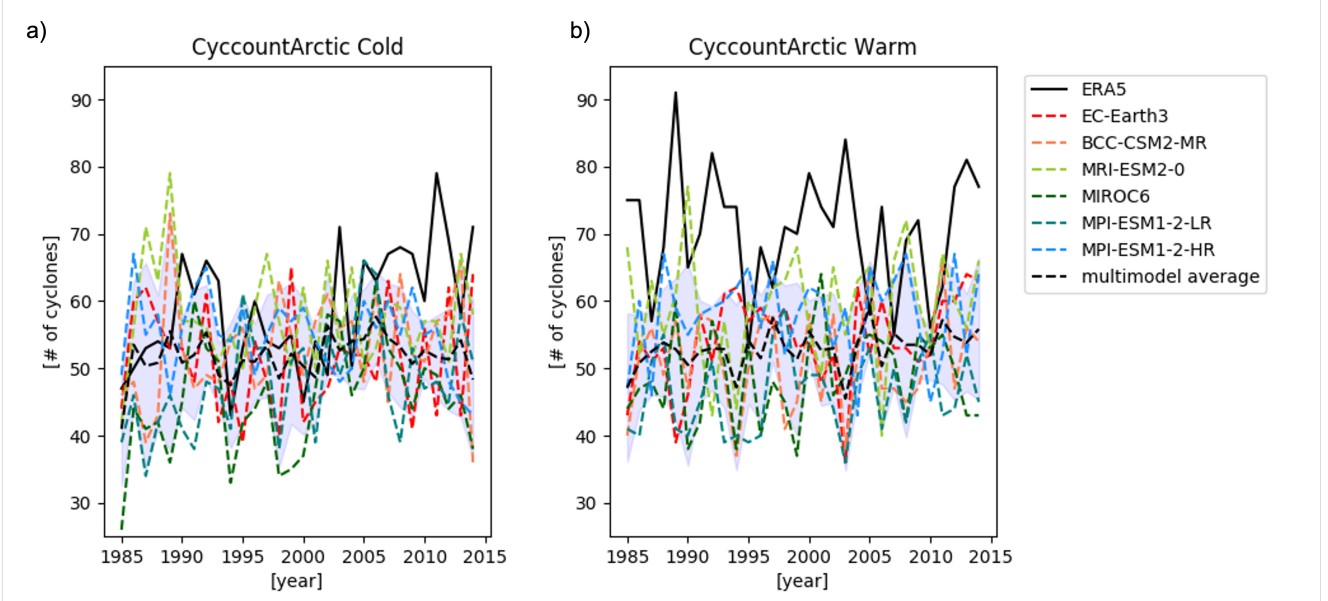

**Figure 3.** The basin-wide seasonal cyclone counts for cold season (panel a) and warm season (panel b).

of skill in the models' ability to reproduce the basin wide warm season cyclone count (Fig. 3). The models depict 51.8 and 52.9 cyclones in the cold and warm seasons respectively, whereas ERA5 depicts 59.0 cyclones in the cold season and 70.5 cyclones in the warm season (Table S1). The reanalysis has a clear seasonal cycle in the cyclone counts with considerably higher counts in the warm season than in the cold season, a signal that is not visible in any of the models as shown in Figure 3. In addition

to underestimating the cyclone counts in the warm season, the models also fail to show the observed cold season trend in the cyclone count data. The ERA5 data shows a significant positive trend (at the 90% confidence level) of cyclone counts, of 3 cyclones/season, in the cold season starting in the early 2000s. This positive trend in the cold season cyclone counts was also observed by Valkonen et al. (2021) and was attributed to the decreasing sea ice during the same time period. None of the models reproduce this observed trend, instead the cold season cyclone counts are consistently underestimated by the model

during the last 10 years of the study period.

Like the spatial distribution of cyclone counts, the modeled cyclogenesis patterns correspond to the reanalysis well (Fig. 4). There are no statistically significant differences between the models and ERA5 grid point average cyclogenesis rates (not shown). In the cold season most cyclones are formed north of Greenland and along the coast of Siberia, whereas in the warm season cyclogenesis takes place relatively uniformly across the entire Arctic Ocean. The models reproduce these patterns well,

although the cyclogenesis counts are underestimated by all the models, especially in the warm season (Fig. 4, panels i-n). The grid point differences between the ERA5 and the model averages are statistically non-significant, but the underestimations are consistent with the basin wide seasonal average cyclogenesis rates shown in Figure 5.





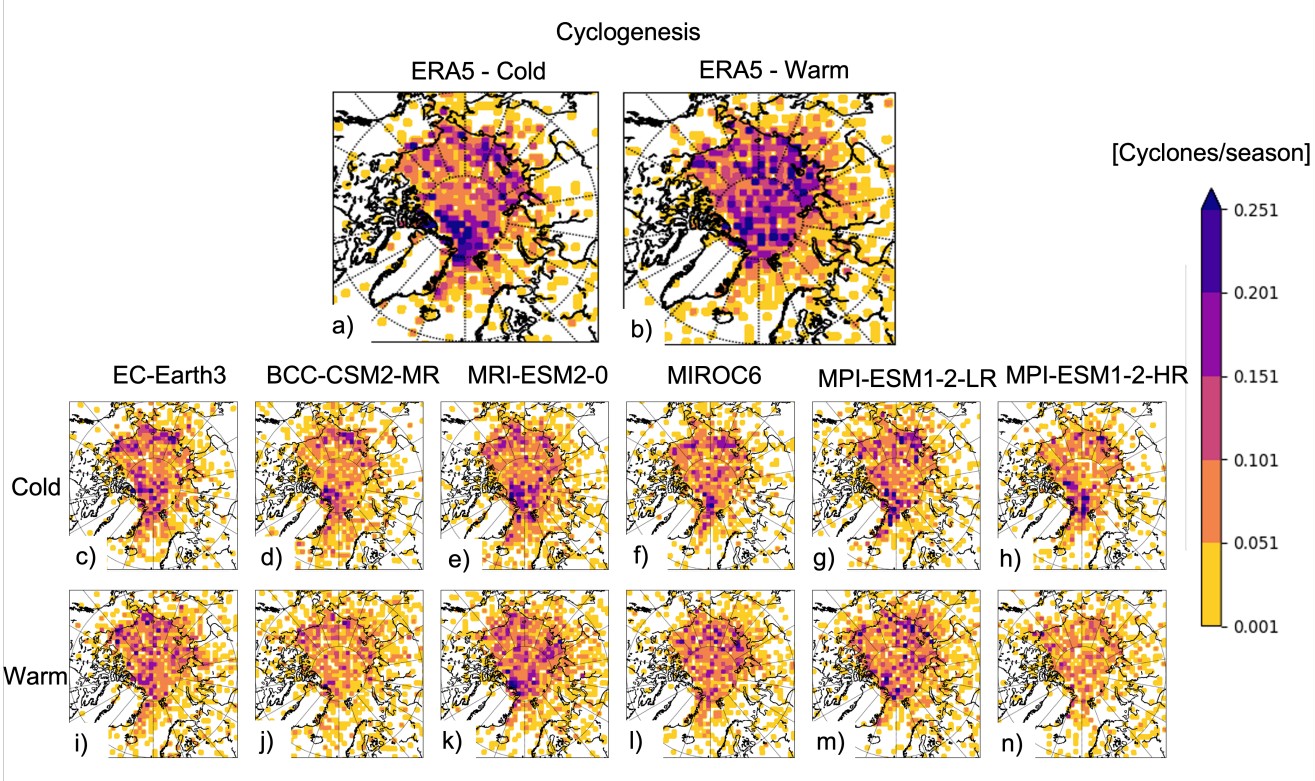

**Figure 4.** The seasonal average cyclogenesis rates for 1985-2014 calculated over 150km x 150km grid boxes for cold and warm seasons.

While the results in Figure 4 show that the spatial genesis patterns are reproduced well by the models, the basin-wide genesis counts are not (Fig. 5). The ERA5 results show an average of 33 genesis events for the cold season and 42 for the warm season (Table S1). The models depict an average of 25 genesis events in both seasons, which is less than ERA5 in both seasons, and the warm season maximum in cyclogenesis rates seen in ERA5 is not replicated by the models. The underestimation of the Arctic cyclogenesis is also prominent in the latter half of the cold season time series (Fig. 5a). There is a positive trend in genesis rates that is observed in ERA5, which matches the positive trend in ERA5 cyclone counts in the cold season (Fig. 3a), which are not represented by any of the models. As mentioned in the previous paragraph, the positive trend in the cyclone counts in the ERA5 data in the last 10 years is associated with the sea ice decline during the same period according to Valkonen et al. (2021). This leads us to believe that the differences in cyclogenesis (and cyclone counts) between ERA5 and the models could be related to the changing sea ice cover and the models' inability to represent the cyclone response to decreasing sea ice cover.

Arctic cyclones can either form within the Arctic (Arctic cyclogenesis) (shown in Fig. 5, panels a-b) or outside the Arctic and traverse into the region. The models are not consistent with ERA5's percentage of cyclones formed from Arctic vs. non-Arctic cyclogenesis in the Arctic region (Fig. 5, panels c-d). In ERA5 57% (59%) of cyclones form in the Arctic in the cold (warm) season, while in the models 48% (47%) form in the Arctic in the cold (warm) season. All the models underestimate



**Figure 5.** The basin-wide seasonal cyclogenesis rate for cold season (panel a) and warm season (panel b). The percentage of cyclones that are generated in the Arctic (compared to outside the Arctic) in cold season (panel c) and warm season (panel d).





the percentage of Arctic genesis in both seasons but do show that roughly half of the simulated cyclones are forming in the Arctic. The 9% (12%) difference between the multi-model mean (MMM) and ERA5 mean Arctic genesis percentages in the cold (warm) season, is mostly due to the models' inaccuracies with the Arctic cyclogenesis rates, indicating that a similar
number of non-Arctic cyclones enter the Arctic in ERA5 and the CMIP6 models. Based on the difference between the total cyclone counts (Fig. 3, Table S1) and Arctic cyclogenesis rates (Fig. 5, panels a-b; Table S1) it is seen that 27 / 26 cyclones (MMM/ERA5) form outside of the Arctic in the cold season and 28 / 29 cyclones form outside of the Arctic in the warm season. Since the number of Arctic cyclones originating outside of the Arctic in the models differs from what is seen in ERA5 by no more than one cyclone/season confirms that the models accurately capture the formation and movement of midlatitude cyclones
into the Arctic. This suggests that the models instead struggle with depicting the correct amount of Arctic cyclogenesis. This underestimation of Arctic cyclogenesis then leads to the underestimation of total Arctic cyclone counts seen in Figure 3. The issues in representing correct local (Arctic) cyclogenesis appear to occur mostly in the warm season and after the last 10 years of the time series in the cold season and may be related to periods of reduced sea ice cover and sea ice melt (3 and 5). Both the warm season in the Arctic and the time period from the 2000s onward in the cold season are times of strong warm season
sea ice melt (9b) resulting in more open water areas and appear to be related to increased cyclogenesis rates in ERA5, but this relationship between ice melt, increased open water and Arctic cyclogenesis is not reproduced by the models.

Next, we move on to discuss cyclone lifetimes and sizes. On average, the cyclones spend between 52 hours in the cold season and 61 hours in the warm season in the Arctic in the ERA5 product (Table S1, Fig. A2) The modeled cyclone lifetime agrees well with the reanalysis and with each other. The multi-model average lifetime is 54 hours in the cold season and 62
hours in the warm season. For all the models and observations, the standard deviation is smaller in the cold season ( 1.3 days) compared to the warm season ( 2 days). In the warm season cyclones tend to spend a longer time in the Arctic, but their lifetime experiences more variability than in the cold season. This means that on average the cyclones that form in the Arctic are longer lived in the warm season than in the cold season and/or the cyclones that traverse to the Arctic in the warm season do so at the earlier stage of their life cycle, which is consistent with the northward shift of the midlatitude storm tracks in the summer. The
fact that the cyclone lifetimes match well between the models and ERA5 indicates that the models are accurately depicting cyclone life cycles.

The geographic distribution of the annual average cyclone radius in ERA5 and the models are shown in Figure A3, while the interannual variability in cyclone radius is shown in Figure 6. The largest cyclones are found outside of the Arctic in the Pacific and North Atlantic storm tracks, with a typical radius of 1600 km. Over the Arctic Ocean, the largest cyclones are
detected in the central Arctic (radius of 1200 km), slightly tilted towards the Siberian and the Chukchi Sea side of the Arctic Ocean, as an extension of the North Atlantic storm track. The modeled basin wide average cyclone radius is 1100 km, while the average radius in ERA5 is 950 km (Table S1, (Fig. 6). The models generally overestimate the cyclone sizes by 15% compared to ERA5 with the models most overestimating cyclone radius near coastal areas, except for BCC-CSM2-MR, for which the overestimation covers most of the Arctic Ocean. The BCC-CSM2-MR has the largest overestimation of cyclone size ( 29%)
year round and is especially pronounced in the warm season. The other models perform better, but still produce too large cyclone radii compared to the observations (Fig. 6 and A3; Table S1). These over estimations are likely related to the lower





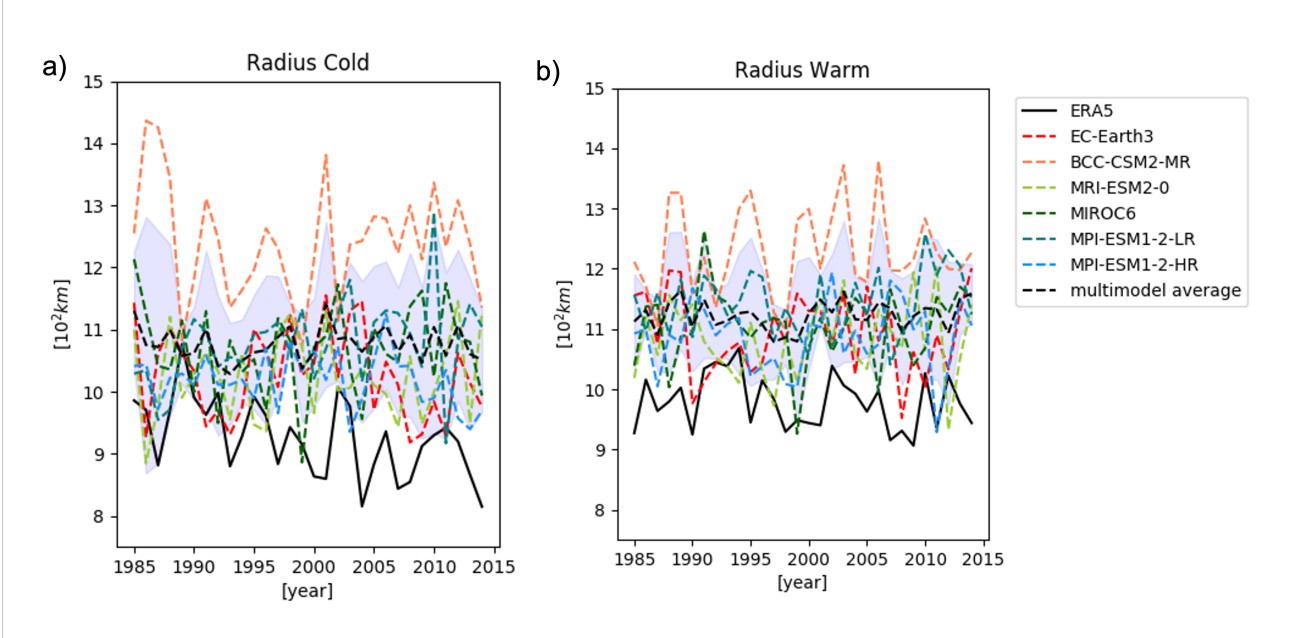

**Figure 6.** The average seasonal cyclone radius (for all the cyclones inside the Arctic) for cold season (panel a) and warm season (panel b).

native spatial resolutions of the models compared to ERA5. The lower native resolutions do not fully explain however, why BCC-CSM2-MR is doing worst of all the models in simulating cyclone sizes, as it does not have the lowest native resolution among the models. Other possible reasons will be discussed in conjunction with the intensity results below.

Interannual variability of cyclone radius is large, but seasonal variability is almost nonexistent (Fig. 6). In the cold season the models also fail to reproduce the consistent statistically significant downward trend in the cyclone sizes (-3.8 km/season) that is present in the ERA5 product (Fig. 6). This is also true for other cyclone characteristics, such as counts (Fig. 3), cyclogenesis (Fig. 5) and intensity (not shown) and follows the pattern of the models failing to depict long-term trends in cyclone characteristics correctly. The same pattern will continue with the cyclone intensity presented next.

The basin wide statistics of the cyclone intensity metrics, which are summarized in Tables 1 and S1, can be seen in Figure 7. The results show that for example, the ERA5 data depicts almost the same mean cyclone DpDr of 81 Pa/km for the cold and warm seasons, but the multi-model mean underestimates the DpDr in both seasons with a statistically significant underestimation of -5.3 Pa/km (-6.3%) in the cold season and -11.4 Pa/km (-13.8%) in the warm season (Table 1, Table S1). The multi-model mean (MMM) depth metric is overestimated by 94.8 Pa (13.9%) in the cold season compared to the mean depth

of 791.4 Pa in ERA5, but shows no statistically significant differences to the ERA5 product in the warm season. The MMM central pressure and ACE metric are underestimated by the models in both seasons. The mean central pressure is 300.8 Pa lower (-0.31%) in the cold season and 82.1 Pa lower (-0.1%) in the warm season compared to the ERA5 mean of 997 hPa (995 hPa) in the cold (warm) season. The MMM ACE is underestimated by 6.9 $m^2s^{-2}$ ( -15.8%) in the cold season and by -8.6





$m^2s^{-2}$ (-20.0%) in the warm season. The differences in the multi-model means are statistically significant (except for depth in
the warm season) at 90 % confidence level. The underestimation of MMM central pressure and overestimation of depth in the
cold season (stronger cyclones) is not consistent with the underestimations of ACE and DpDr metrics (weaker cyclones).

**Table 1.** The difference and percentage difference compared to ERA5 for multi-model mean intensity. Statistically significant differences at
90% confidence level are bolded based on the data on Table S1.

| Intensity metric | Cold season | Warm season |
| --- | --- | --- |
| ACE | **-6.9 m$^2$s$^{-2}$ / -15.82%** | **-8.6 m$^2$s$^{-2}$ / -19.92%** |
| Central pressure | **-300.8 Pa / -0.30%** | **-82.1 Pa / -0.08%** |
| Depth | **94.8 Pa / 13.87%** | 3.7 Pa / 1.08% |
| DpDr | **-5.3 Pakm$^{-1}$ / -6.34%** | **-11.4 Pakm$^{-1}$ / -13.83%** |

We begin the discussion of possible reasons for the differences between the different intensity metrics with the three pressure
based metrics and lastly will discuss ACE with respect to the other metrics. The underestimation of central pressure metric
compared to ERA5 product (meaning stronger cyclones), is related to low biases in the models' SLP fields as shown by Figure
8. Models that show large and statistically significant underestimation (overestimation) of SLP in the Arctic region also show
a general underestimation (overestimation) of cyclone central pressure. Models with no statistically significant differences in
spatial SLP fields show no statistically significant differences in cyclone central pressure.

The results of cyclone depth also show overestimation of cyclone strengths by the models, with a MMM difference of 94.8
Pa (13.9%) in the cold season and 3.7 Pa (1.1%) in the warm season. As seen in Figure 8 the central pressure differences
between the CMIP6 models and ERA5 is in all but one case (HR in the cold season) is larger than the depth difference. This
result is consistent with the discussion in the previous paragraph, that overall biases in SLP in the models, lead to an offset
in central pressure between the models and ERA5. But, since depth also differs between the models and ERA5, and by a
smaller magnitude than central pressure, we can conclude that the pressure biases are not uniform across the cyclones and
that mechanisms other than a simple difference in SLP between models and ERA5 is present that alters the strength of Arctic
cyclones in the CMIP6 models. To gain insight into the source of the depth differences the intensity of cyclones originating
within and outside of the Arctic was explored.

The results (Table 2, Fig. A4) show us that in general the depth for cyclones that form outside of the Arctic is overestimated
more than for cyclones that form inside the Arctic. In the cold season cyclones that formed in the Arctic were modeled more
accurately ( 19.8 Pa / 5.9 % difference to ERA5), than cyclones that formed outside of the Arctic ( 34.8 Pa / 16.4 % difference
to ERA5) (Table 2). In the warm season the Arctic formed cyclones were actually underestimated by the models (-34.3Pa /
-4% difference to ERA5) whereas the cyclones formed outside the Arctic were still overestimated compared to ERA5 depths
(15.6 Pa / 2.8% difference to ERA5). This difference in cyclone depth errors between cyclones that form within or outside of
the Arctic is important because as discussed above, Arctic cyclogenesis rates are underestimated in all of the models and thus
a larger fraction of the cyclones that are in the Arctic in the models are coming from midlatitudes compared to in the ERA5
product (Fig. 5 panels c and d). Since these non-Arctic formed cyclones are stronger than their Arctic formed counterparts,





**Figure 7.** The difference of mean cyclone intensity compared to ERA5 mean in the Arctic (within the study region) for 1985-2014 for each model separated. The multi-model mean (MMM) is shown first in all panels.





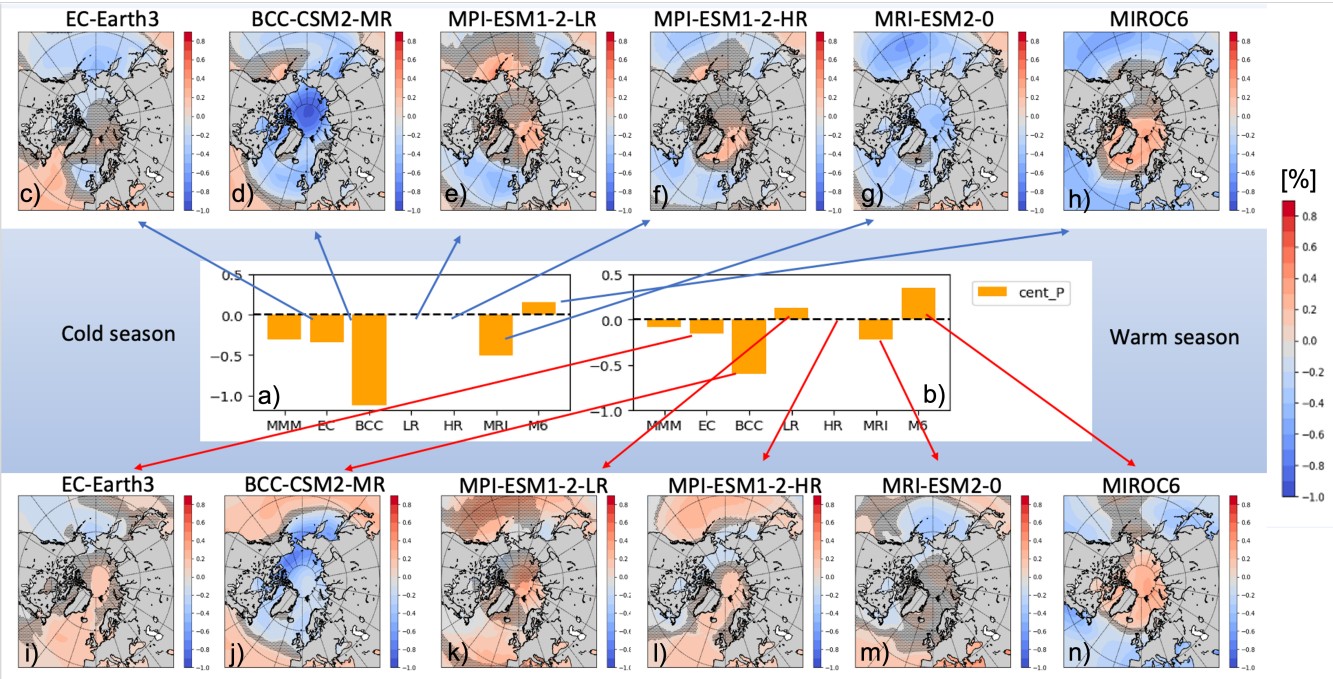

**Figure 8.** The percentage difference of SLP between each of the CMIP6 models and ERA5 (the contour plots) for cold (top row) and warm (bottom row) seasons alongside the the percentage differences in cyclone central pressure between each model and ERA5. The stippling in the SLP panels shows areas of statistically non-significant areas at 90% confidence level

**Table 2.** The percentage difference compared to ERA5 for multi-model mean intensity for cyclones generated in the Arctic and outside the Arctic. All differences shown are statistically significant at 90% confidence interval, except 0% differences.

| Intensity metric | Genesis location | Cold season | Warm season |
|---|---|---|---|
| ACE | Arctic | -8.7 $m^2s^{-2}$ / -22.3% | -9.9 $m^2s^{-2}$ / -24.2% |
|  | Outside | -6.9 $m^2s^{-2}$ / -13.9% | -8.7 $m^2s^{-2}$ / -18.1% |
| Central pressure | Arctic | -154.9 Pa / 0% | -5.6 Pa / 0% |
|  | Outside | **-365.9 Pa / -0.4%** | **-77.1 Pa / -0.1%** |
| Depth | Arctic | **19.8 Pa / 5.9%** | **-34.3 Pa / -4%** |
|  | Outside | **34.8 Pa / 16.4%** | **15.6 Pa / 2.8%** |
| DpDr | Arctic | **-7.4 Pakm$^{-1}$ / -9.5%** | **-13.7 Pakm$^{-1}$ / -17.2%** |
|  | Outside | **-4.8 Pakm$^{-1}$ / -4.2%** | **-11.4 Pakm$^{-1}$ / -13.1%** |

the modeled central pressure is underestimated and the depth is overestimated compared to ERA5. These results show how errors in the location of cyclogenesis and/or errors that result in too few Arctic cyclogenesis cases contribute to errors in Arctic





cyclone intensity. Other metrics were also affected by the cyclogenesis location, but the results were not as consistent as with the depth metric.

The final pressure-based metric, DpDr, is almost always underestimated by the models, in contrast to depth which is almost always overestimated by the models. This inconsistency in the sign of the intensity difference between depth and DpDr is likely related to errors in the size of the cyclones. As shown in Figure 6 the models consistently overestimate the size of Arctic cyclones compared to ERA5. As a result, even though most models simulate cyclones that are deeper than in ERA5 this greater depth occurs over a larger distance resulting in a weaker pressure gradient (DpDr) across the cyclone. The multi-model

ensemble DpDr is -5.3 Pa/km( -6.3%) in the cold season and -11.4 Pa/km (-13.8%) in the warm season compared to the ERA5. These large differences are due to two competing factors. The seasonal variability of DpDr shows the opposite seasonal cycle to depth and central pressure differences, with larger differences in the warm season. MMM cyclone depth shows no statistically significant differences to ERA5 in the warm season, whereas the cyclone radius is overestimated by all the models in both seasons leading to a large underestimation of DpDr in the warm season.

Next, we will consider the kinetic-energy based intensity metric, ACE. The ACE metric is underestimated by most of the models with a MMM ACE of -6.9 $m^2s^{-2}$ (-15.8%) in the cold season and -8.6$m^2s^{-2}$ (-19.9%) in the warm season compared to the ERA5 mean. These large differences are due to an underestimation of the average wind speed within the cyclones, which we attribute, in part, to an underestimation of DpDr in most models. However, the DpDr underestimation does not fully explain the large underestimation of ACE metrics. As shown in Valkonen et al. (2021) the ACE intensity metric can be influenced by

differences in the surface roughness of the ocean or sea ice and as discussed below it appears that the different representation of surface roughness in the models is indeed influencing the ACE in the models. In addition to differences in sea ice or ocean roughness between the models and ERA5 differences in the amount of sea ice present in the models will also alter ACE, relative to that in ERA5, since sea ice has a greater roughness than open water.

In both seasons the BCC-CSM2-MR, MIROC6 and both MPI models underestimate ACE, with the largest underestimation

in BCC-CSM2-MR (  -30 m2s2 / -80%) followed by MIROC6 (  -15 $m^2s^{-2}$ / -35%), MPI-ESM1-2-LR (-6.1 $m^2s^{-2}$ / -14% in the cold season and -10 $m^2s^{-2}$ / -20% in the warm season) and lastly MPI-ESM1-2-HR with -1.9 $m^2s^{-2}$ / -5% (-4.9 $m^2s^{-2}$ / -10%) in the cold (warm) season. All these models underestimate DpDr relative to ERA5 and also have a very different surface roughness from ERA5. In ERA5 the sea ice surface roughness of sea ice varies between 0.001 m and 0.006 m (ECMWF, 2006, 2016; L. Jakobson et al., 2019). The BCC-CSM2-MR model uses constant sea ice surface roughness length of 0.001 m, when

surface temperature is less than -2°C, but the roughness length increases to 0.008 m, when surface temperatures are higher than -2°C. The surface roughness in MIROC6 is 0.01 m and 0.001 m in MPI-ESM1-2 models (Tandon et al., 2008). These different surface roughness values are mostly consistent with ACE differences in Figure 7. MIROC6 and BCC-CSM2-MR, in the warm season when surface temperature is mostly above -2°C, have larger roughness lengths than ERA5, which together with the underestimation of DpDr leads to lower ACE values in these models compared to ERA5. In the cold season, the

surface roughness over ice is mostly lower than in ERA5, but the BCC-CSM2-MR has more sea ice in the Central Arctic, which together with the lower DpDr leads to underestimation of ACE. In contrast to the three models just discussed, which underestimate DpDr, MRI-ESM2-0 model shows accurate DpDr in the cold season and a slight underestimation in the warm





season, but the ACE metric is overestimated by  37% (17.1 $m^2s^{-2}$ in the cold season and 13.5 $m^2s^{-2}$ in the warm season) compared to ERA5. MRI-ESM2-0 has a constant surface roughness of 0.001 m (Yukimoto et al., 2013) that is equal to the

lower bound of ERA5 surface roughness. This means that for most of the time the surface roughness in MRI-ESM2-0 is less than in ERA5, which leads to the overestimation of ACE, even though DpDr is nearly the same or slightly underestimated. Of the 6 models evaluated here EC-Earth3 has the smallest ACE errors relative to ERA5 with almost no difference in the cold season and slight underestimation of -2.7 $m^2s^{-2}$ (-5.7%) in the warm season. The small ACE errors in EC-Earth3 are due to the combination of both small DpDr errors as well as EC-Earth3 using the same surface roughness as ERA5 since EC-Earth3

and ERA5 are based on the same model core.

## 3.2   Sea Ice

Next, we move on to assessing the changes that take place in Arctic sea ice and how well the models represent these changes. Arctic cyclones and the surface are strongly intertwined, so the SIC plays a large part in the interactions between the cyclones and the surface. In addition, the strong warming in the Arctic can change these interactions as sea ice declines, which is why it

is important of understand how well the models depict the sea ice and its interactions with cyclones.

Average seasonal SIC encountered by the cyclones is calculated as follows. For each cyclone in the study area, the average SIC within the cyclone area is calculated. This cyclone area SIC average is assigned to the central cyclone position and is averaged for all cyclones at this location in each season (Figure A13). This cyclone area average SIC is not identical to the climatological SIC but is weighted by areas of high cyclone activity and influenced by the cyclone area such that each grid point

in Figure A13 represents the average SIC across the entire area of all cyclones at that point. The SIC varies between 70%-100% in the cold season and 40% - 100% in the warm season. The spatial patterns of SIC varies from covering the fully frozen Arctic Ocean in the cold season to the more melted warm season Arctic, where the >90% SIC is located north of Greenland and the Canadian Archipelago, with the Alaskan and Siberian coastal regions and the Barents and Kara seas having lower SIC. These results highlight the different lower boundary condition the cyclones experience between the cold and warm season with most

of the cyclone area having consolidated ice cover in the cold season but a large fraction of the cyclone area being loose ice or open water in the warm season.

The multi-model basin-wide mean SIC is underestimated in both the cold and warm seasons by the models (Fig. 9), but the models are able to replicate the strong negative sea ice trend in the warm season with a high degree of accuracy. The average MMM SIC is 92% (93% for ERA5) for the cold season and 68% (73% for ERA5) for the warm season. The observed trend in

ERA5 is 0% per season in the cold season and -1% per season in the warm season, whereas the MMM is 0% per season and -0.4% per season in the cold and warm seasons respectively. The BCC-CSM2-MR is consistent with the observed SIC values in both seasons. Both MPI-ESM1-2 models underestimate the SIC in both seasons, with stronger underestimation in the warm season. MRI-ESM2-0 overestimates the SIC in the cold season but is consistent with the observations in the warm season, whereas MIROC6 underestimates SIC in the cold season, but overestimates SIC in the warm season. EC-Earth3 is consistent

with the ERA5 in the cold season, but overestimates SIC in the warm season.





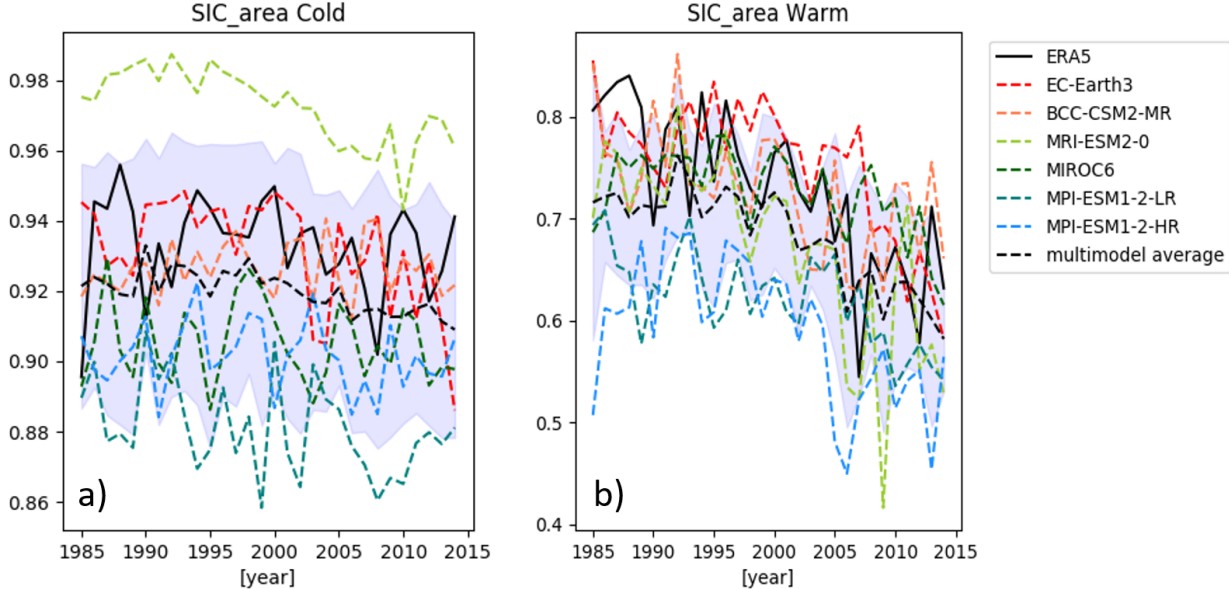

**Figure 9.** The basin-wide seasonal cyclone SIC for cold season (panel a) and warm season (panel b).

While the SIC trends are modeled correctly (mostly), the models do not replicate the relationship between changes in SIC and Arctic cyclogenesis rate seen in ERA5 (Figure 5). For example, cyclogenesis in the cold season should be increasing in association with the strong decline in the warm season SIC to be consistent with ERA5, but the modeled cyclogenesis rates (Fig. 5 panels a and b) stay constant throughout the study period.

### 3.3    Relationships Between Cyclone Counts and Sea Ice

The cyclone characteristics and sea ice results above have shown that the models do a reasonable job representing the SIC trends but do less well with the average SIC and cyclone characteristics, especially with representing interannual variability and trends correctly. To gain additional insight into how the relationships between cyclones and sea ice is represented in the models, trend matrices (Fig. 10) were compared between the CMIP6 models and ERA5 for the cold season cyclone counts (Figure 10, panels a and c-h) and warm season SICs (Figure 10, panels b and i-n). The trend matrices show the statistically significant trends (blue for positive sea ice trends, red for positive cyclone count trends) for the respective start year (x-axis) and time range over, which the trend is calculated for (y-axis). This allows us to get more robust understanding of the trends and interannual variability in the models rather than simply focusing on a single trend over the whole study period.

What is seen in the trend matrices for ERA5 in Figure 10 is consistent with the findings in Valkonen et al. (2021) who found an increasingly positive trend in cyclone counts throughout the study period, with enhanced positivity from late 1990s onward (Fig. 10a). This increase in cyclone counts is associated with an increasingly negative trend in the ERA5 SIC (Fig. 10b). It



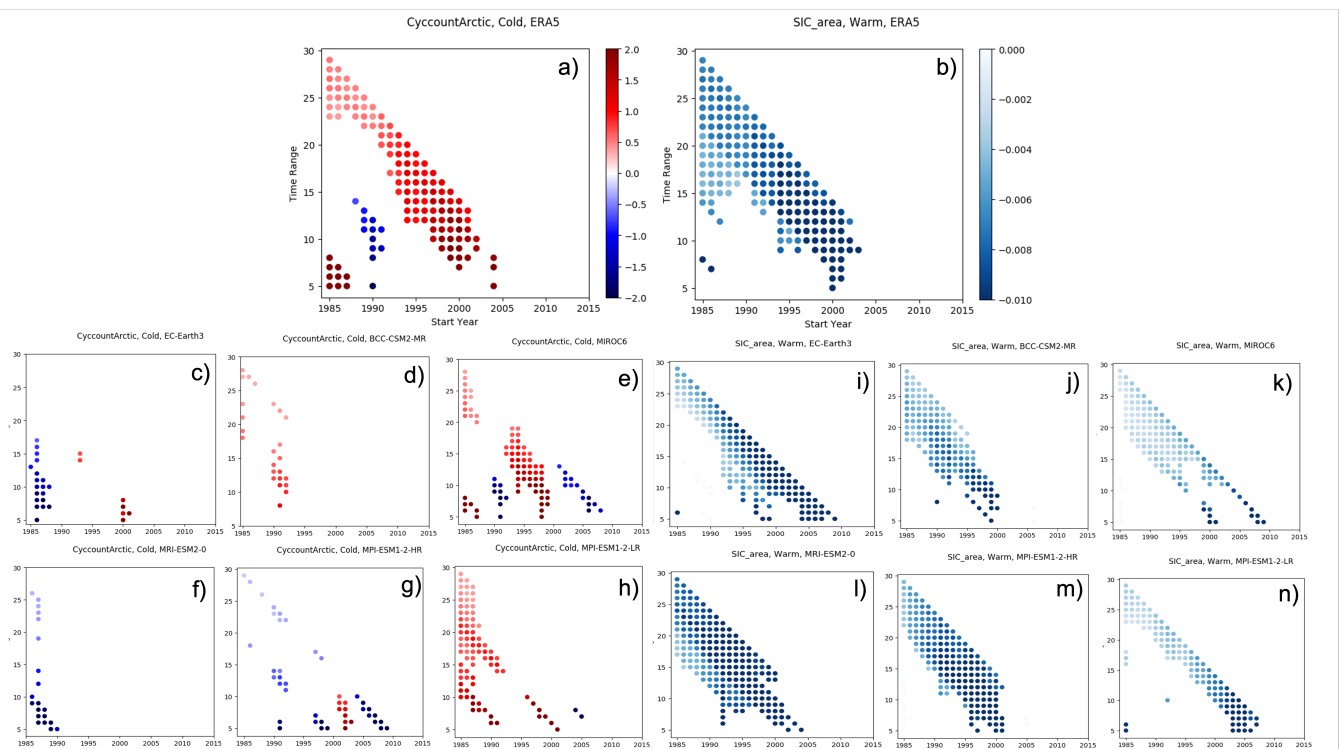

**Figure 10.** Trendmatrices for cyclone counts and SIC

is noteworthy however, that similar relationships between cyclone count and SIC trends is not found in any of the models. None of the CMIP6 models reproduce the cyclone count trend patterns (Fig. 10, panels c – h), despite the models generally reproducing the SIC changes (Fig. 10, panels i – n) relatively well. As shown earlier (Fig. 5) errors in cyclone counts in the CMIP6 models are largely related to the models inability to reproduce the observed rate of Arctic cyclogenesis. This along with the cyclone count and SIC trend results (Fig. 10) lead us to believe that the reason behind the underestimation of cyclone counts over the Arctic in CMIP6 models is because the models are not fully presenting the relationships between the sea ice decline and Arctic cyclogenesis.

Lead-lag correlation analysis was also performed between cyclone counts, SIC and NAO in order to assess if differences in large-scale circulation, as represented by the NAO, are related to the cyclone count errors. These results are shown in the supplemental materials (Fig. A14). The lead-lag analysis showed strong correlation between cyclone counts and SIC in the ERA5 data, with weaker relationship between the cyclone counts and NAO. The cyclone count-SIC correlations in ERA5 were stronger when lagged, suggesting a positive feedback loop between the increased cyclone counts and SIC, where less sea ice is related to higher cold season cyclone counts and higher cyclone counts result in reduced SIC in both seasons. In contrast for the CMIP6 models SIC was not significantly correlated with the cyclone counts in any season or lag; whereas NAO was correlated





with the cyclone counts. It appears that Arctic cyclones in the CMIP6 models are more strongly modulated by the large-scale circulation than local processes, with the positive cyclone counts-SIC feedback loop found in the reanalysis products missing.

## 4    Discussion and Conclusions

In this study we examined how cyclones over the Arctic Ocean and their relationship with sea ice is represented in an ensemble
of 6 different historical simulations from the suite of CMIP6 models. A cyclone tracking algorithm was applied to the models'
sea level pressure which resulted in a cyclone climatology that included multiple cyclone characteristics such as cyclone
counts, duration, size and intensity. These characteristics were then compared to the cyclone climatology derived from ERA5
to evaluate how well the CMIP6 models simulated Arctic cyclones, their characteristics and their relationships to the sea ice.

The spatial patterns of seasonal cyclone counts and cyclogenesis (Figs. 2, 3, and A1) were well replicated with only slight
(mostly statistically non-significant) underestimation of the counts and genesis rates in both seasons. However, the basin-
wide multi-model averages depicted larger differences (Figs. 3 and 5). In the cold season all the models produce mostly
reliable results at the beginning of the study period but failed to reproduce the observed increase in cyclone counts from early
2000s onward (12.3% underestimation compared to ERA5 in the last ten years). In the warm season the models consistently
underestimate the cyclone counts with an average difference of 25% difference compared with ERA5. This underestimation
of Arctic cyclone counts was shown to be due to underestimation of Arctic cyclogenesis (Fig. 5) (underestimation of -25%
in the cold season and -40% in the warm season), with stronger underestimation in times of reduced sea ice (summer and the
later portion of the analyzed time period in the cold season). Part of this underestimation could be internal variability in the
models. However, Valkonen et al. (2021) as well as results in Figure 10 and supplemental material (Figure A14) have shown
that cyclone counts in ERA5 are correlated negatively with sea ice changes leading to more cyclones as sea ice declines. The
models do not reproduce the lead-lag correlations or the trend patterns in cyclone counts seen in ERA5 (Fig. A14, 10). As
shown by the basin-wide SIC, cyclone count and cyclogenesis rate results (Figs. 9, 3 and 5 and ) the models are able to mostly
reproduce the sea ice decline consistently with ERA5 in both seasons, but fail to reproduce the correct cyclogenesis rates (and
hence cyclone counts) in either season. This leads us to believe that, while the models are doing well reproducing the general
patterns of SIC and cyclone characteristics, they struggle to reproduce the local processes related to the sea ice decline and its
influence on cyclone development in the Arctic.

Cyclone size was another characteristic investigated in this study 6. We found that all of the models struggled capturing
cyclone radii relative to ERA5. In both seasons the models overestimated cyclone size with the BCC-CSM2-MR model dras-
tically overestimating cyclone radius. This is mostly due to the considerably lower resolution in the CMIP6 models compared
to the ERA5 product. However, the lower horizontal resolution does not alone explain the large underestimation of cyclone
sizes by the BCC-CSM2-MR model, which has the same resolution as EC-Earth3 and MRI-ESM2-0 models. Additionally,
ERA5 shows a downward trend in cyclone size with time that neither the MMM nor any of the individual models capture. This
overestimation was found to influence cyclone intensities measured by the DpDr metric, as the consistent overestimation led
to a consistent underestimation of the DpDr in all the models (Fig. 7).





The models were also found to both underestimate or overestimate the cyclone intensities, compared to ERA5 product,
depending on the metric chosen and model considered (Figs. 7, 8, A4 and Tables 1,2 and S1). The central pressure and depth
metrics (only in the warm season for depth) overestimated cyclone strength, whereas ACE and DpDr underestimated cyclone
strength based on the multi-model means. These inconsistencies between the metrics were at least partly attributed to the
unique characteristics of each of the individual metrics. As the ACE metric is calculated based on near surface mean wind
speeds, it is strongly tied to the surface roughness used in each of the models. Central pressure and depth on the other hand
were found to be strongly coupled and consistent with respect to each other and showed relationships to broader patterns of
SLP biases in the models. Models that underestimated central pressure also overestimated depth (Figs. 8, A8 and A10). This
makes sense, as depth is calculated based on the central pressure. Overestimation of cyclone depths and central pressure in the
cold season were suggested to be at least partly associated with the cyclogenesis location of the cyclone. Cyclones that formed
in the Arctic were closer to the ERA5 depths (5.9% difference) than cyclones that formed over midlatitudes and moved into
the Arctic (16.4% difference) (Fig. A4, Table 2). DpDr is calculated based on depth and cyclone size, and as all the models
consistently overestimate the cyclone radii (Fig. 6 Table S1), this leads to an underestimation (in most models and both seasons)
of the DpDr.

In addition, model resolution also plays a role in intensity biases, as depicted by the MPI-ESM1-2-LR and MPI-ESM1-2-
HR models. The two models have same underlying physics and dynamical core, but different resolutions, hence any resulting
output differences are due to the differences in the nominal resolutions. The higher resolution MPI-ESM1-2-HR model shows
smaller differences in cyclone intensities between the model results and ERA5 than the MPI-ESM1-2-LR model (Fig. 8, Table
3). Also, the model resolution appears to influence the intensity of cyclones formed in the Arctic more than the once formed
outside the Arctic (Table 2). This could be because, as the model resolution decreases some of the smaller, more intense
mesoscale cyclones are missed by the models, causing the intensity distribution to be underestimated.

Model resolution is also playing a role in cyclone counts, genesis, and sizes, with models with higher resolution being more
consistent with the ERA5 results in general (Table S1). For example, MIROC6 has the lowest nominal resolution with 250 km
horizontal resolution. The MIROC6 displays consistent underestimation of cyclone strengths in both seasons; lower cyclone
counts and Arctic cyclogenesis than most of the other models as well as generally larger cyclones. This is all consistent with
lower resolution models missing some of the smaller cyclones, as shown for example by Valkonen et al (2021) for different
resolution reanalysis products and with the generally poorer skill of lower resolution models in depicting Arctic cyclones
(Song, et al., 2021).

The correct modeling of the interaction between cyclone characteristics in the Arctic and SIC is vital in understanding the
future climate system. Even though the current CMIP6 models are able to reproduce the SIC patterns reasonably well as shown
by the small differences in the SIC trends between the ERA5 and the multi-model mean (Fig. 9, Table S1), they are lacking
in representing the local influence of the Arctic sea ice decline to cyclone characteristics, as shown by the underestimation
of Arctic cyclone counts and genesis rates (Figs. 3 and 5) and the trendmatrix results shown in Figure 10. Instead, they are
overemphasizing the effect of general circulation (NAO) on the Arctic cyclones (Fig A14). This leads to underestimation of
cyclone counts and biases in their intensities, as cyclones originating in the midlatitudes are inaccurately dominating the number





**Table 3.** Warm season percentage differences between the high and low resolution MPI-ESM1-2 models for cyclone intensities for cyclones generated inside the Arctic and outside the Arctic.

|  | MPI-ESM1-2-HR | | MPI-ESM1-2-LR | |
|---|---|---|---|---|
|  | Arctic | Global | Arctic | Global |
| ACE | -13.3% | -11.4% | -29.2% | -21.6% |
| Central pressure | 0% | 0% | 0.22% | 0.12% |
| Depth | 0% | 0% | -15.2% | 0% |
| DpDr | -14.4% | -12.0% | -25.1% | -21.3% |

of Arctic cyclones. Individual models also have issues that result in misrepresentation of other cyclone characteristics, such as
cyclone radius and intensity. These include for example, low nominal resolution of the models (MIROC6, MPI-ESM1-2-LR), differences in surface roughness schemes and surface temperature biases in the Arctic (BCC-CSM2-MR).

In the future more research is needed in developing these models to correctly reproduce the affects SIC decline has on the various cyclone characteristics. As more CMIP6 data becomes available, repeating this study with a larger model ensemble, and possibly excluding outlier models would increase the reliability of these findings.

*Data availability.* The CMIP6 model output is available through the ESGF website, and ERA5 data through Copernicus. The Arctic Cyclone Catalogs that were created from the tracking algorithm output are available through NSF Arctic Data Center.

*Author contributions.* EV planned the study with substantial inputs from JC and EC. EV and MS retrieved and pre-processed the CMIP6 data for analayses. EV performed the analysis and wrote the manuscript with reviews and edits from JC and EC. EC uploaded the cyclone catalogs to NSF Arctic data center. JC acquired the financial support for the project leading to this publication.

*Competing interests.* The authors declare no competing interests.

*Acknowledgements.* This work was supported by National Science Foundation award PLR 1603384. This work utilized the Summit supercomputer, which is supported by the National Science Foundation (awards ACI-1532235 and ACI-1532236), the University of Colorado Boulder, and Colorado State University. The Summit supercomputer is a joint effort of the University of Colorado Boulder and Colorado State University.



**Appendix A:  Additional Tables and Figures**



**Figure A1.** The seasonal average cyclone count for 1985-2014 calculated over 150km x 150km grid boxes for cold and warm seasons. The ERA5 panels show the observed spatial distributions, and the model panels display the statistically significant percentage differences to the ERA5 product at 90 % confidence interval


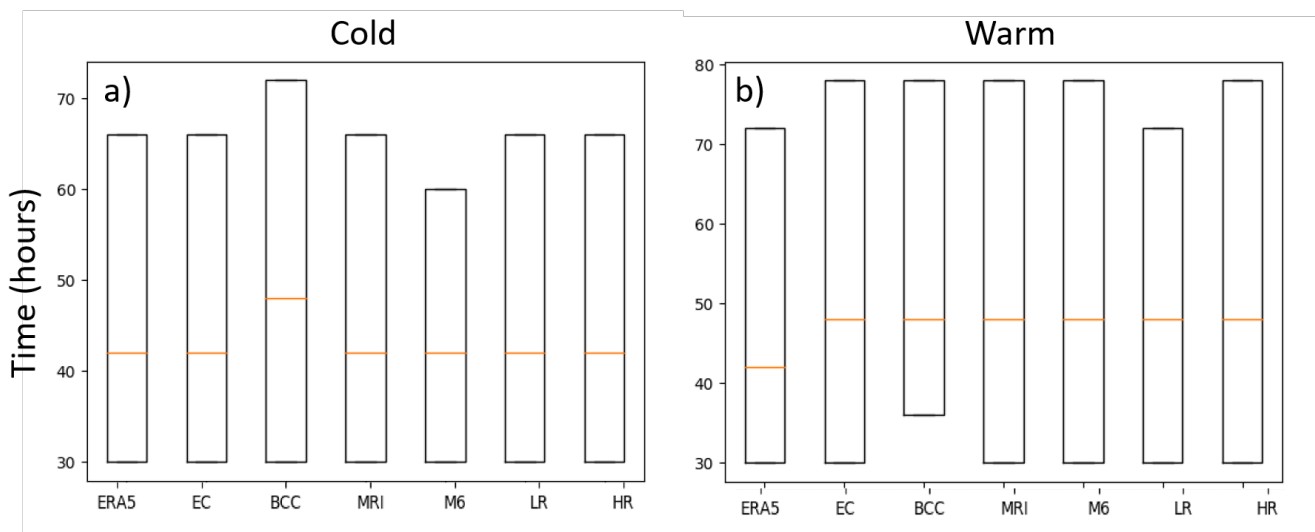

**Figure A2.** Average cyclone lifetime in the Arctic (within the study region) for 1985- 2014 for each model. Boxes extend through the interquartile range, with median shown as a line

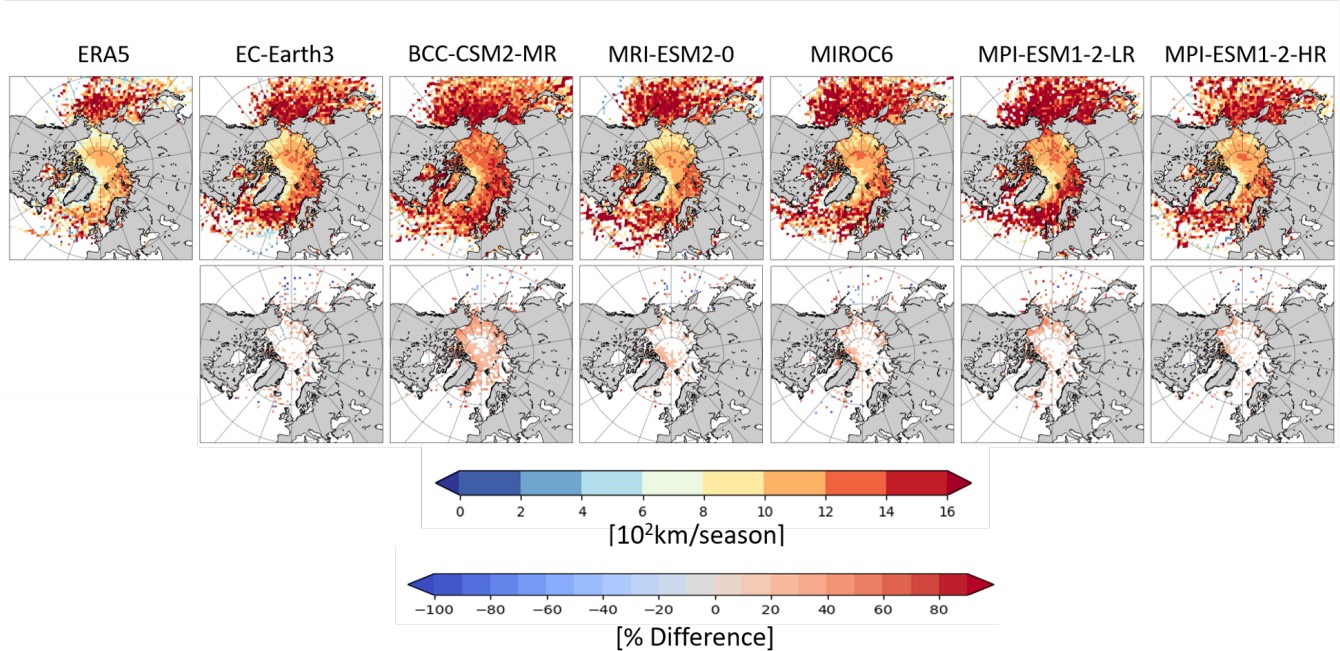

**Figure A3.** The yearly average cyclone radius for 1985-2014 calculated over 150km x 150km grid boxes for cold and warm seasons (top row).The percentage difference between the models and ERA5 for yearly average cyclone radius for 1985-2014 calculated over 150km x 150km grid boxes (bottom row). The statistically significant differences at 90% confidence level are shown.

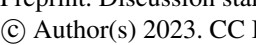



**Figure A4.** The percentage difference of mean cyclone intensity compared to ERA5 mean in the Arctic (within the study region) for 1985-2014 for each model separated for cyclones generated in the Arctic (solid line) and outside the Arctic (dashed line). The multi-model mean (MMM) is shown first in all panels.





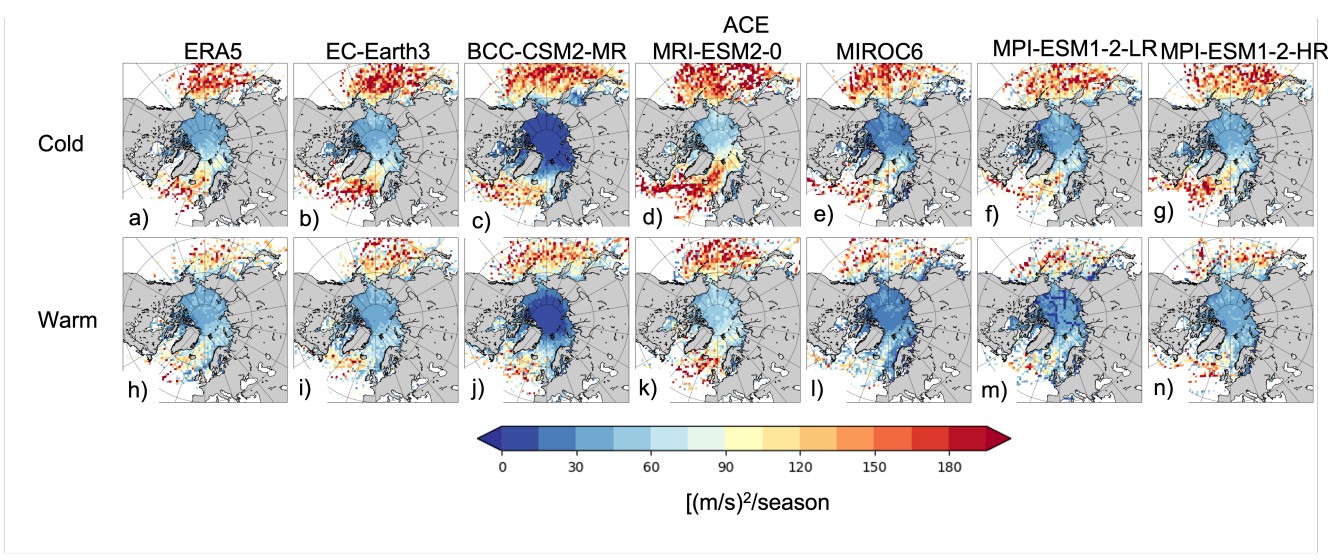

**Figure A5.** The seasonal average cyclone ACE for 1985-2014 calculated over 150km x 150km grid boxes for cold and warm seasons.

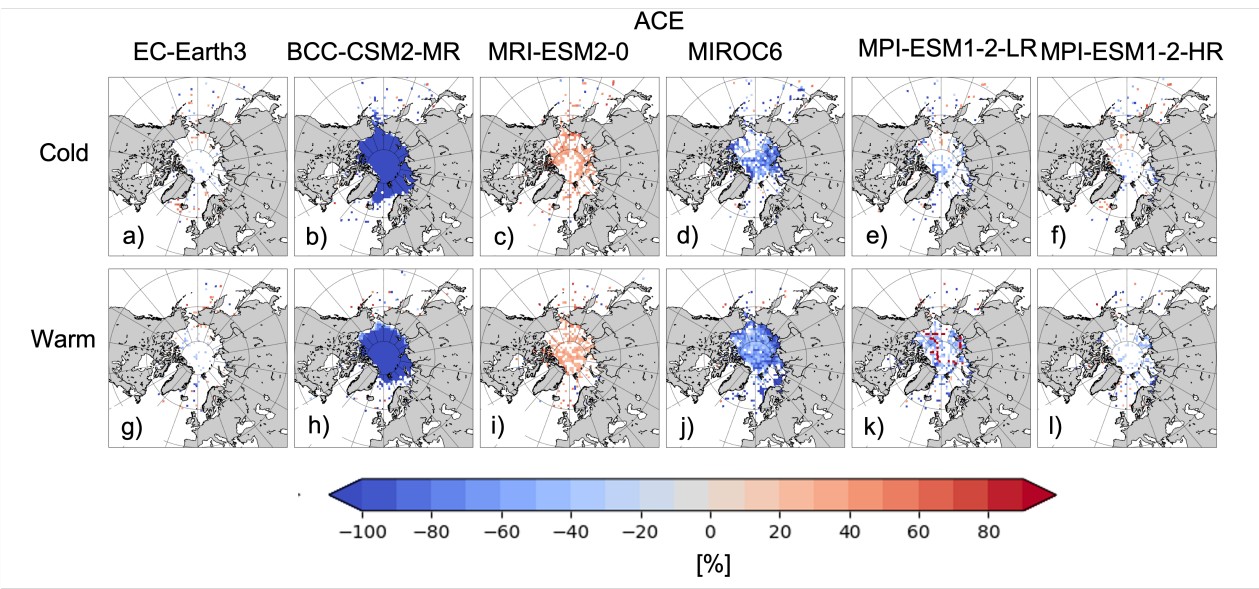

**Figure A6.** The percentage difference between the models and ERA5 for seasonal average ACE for 1985-2014 calculated over 150km x 150km grid boxes for cold and warm seasons. The statistically significant differences at 90% confidence level are shown.





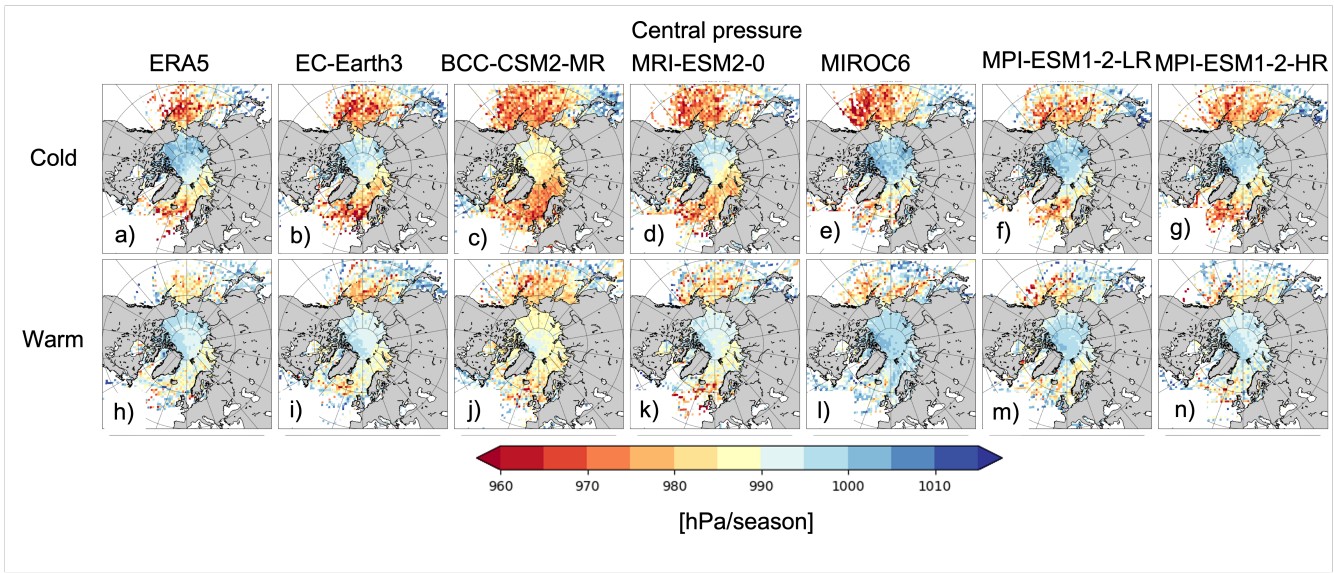

**Figure A7.** The seasonal average cyclone central pressure for 1985-2014 calculated over 150km x 150km grid boxes for cold and warm seasons.

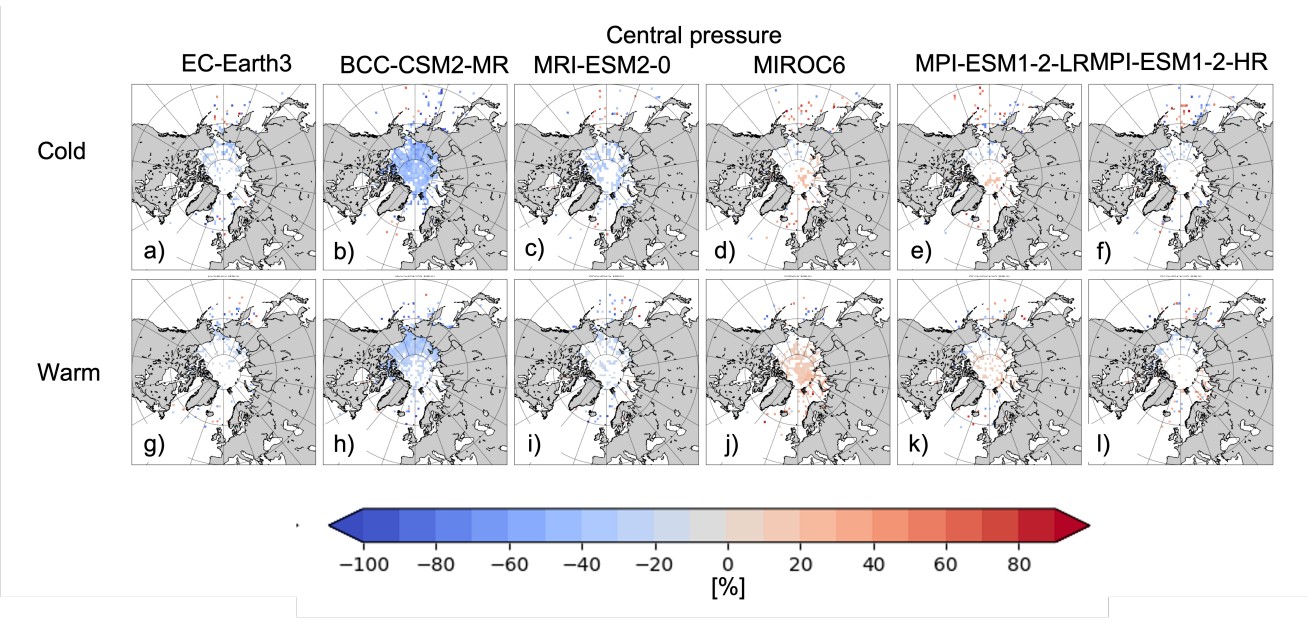

**Figure A8.** The percentage difference between the models and ERA5 for seasonal average central pressure for 1985-2014 calculated over 150km x 150km grid boxes for cold and warm seasons. The statistically significant differences at 90% confidence level are shown.




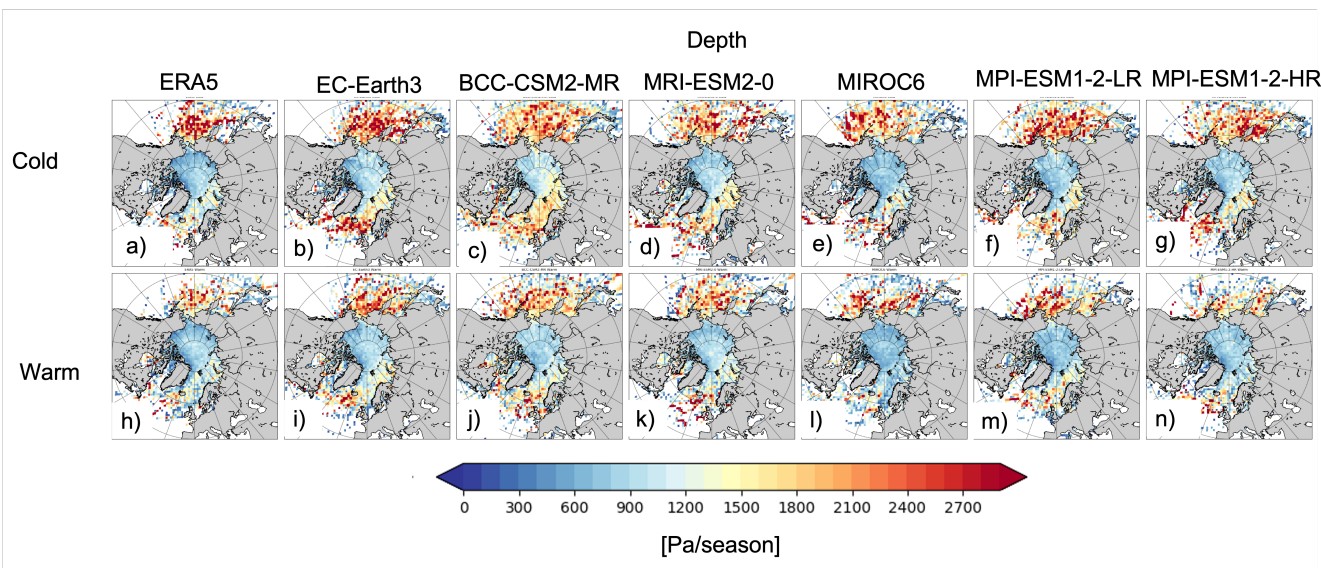

**Figure A9.** The seasonal average cyclone depth for 1985-2014 calculated over 150km x 150km grid boxes for cold and warm seasons.

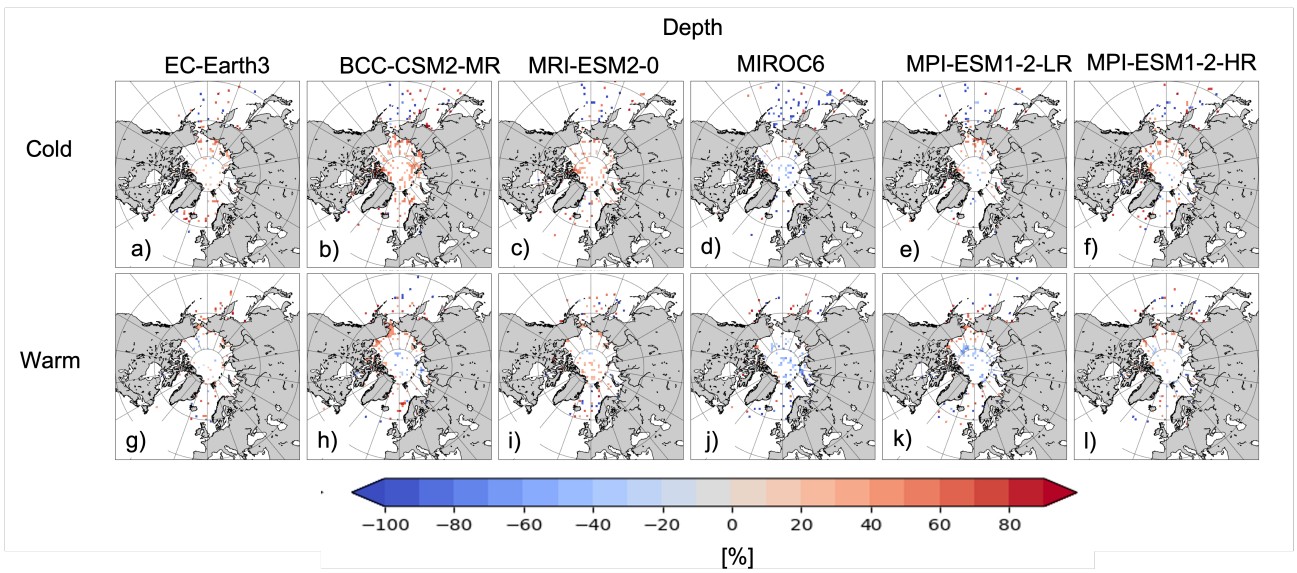

**Figure A10.** The percentage difference between the models and ERA5 for seasonal average depth for 1985-2014 calculated over 150km x 150km grid boxes for cold and warm seasons. The statistically significant differences at 90% confidence level are shown.





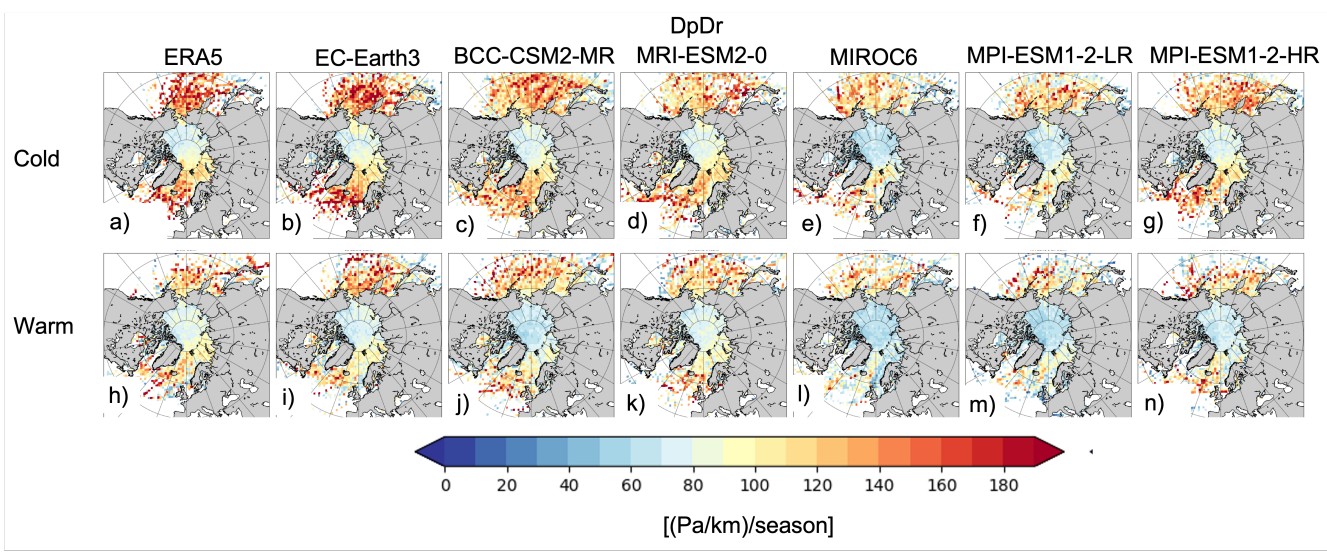

**Figure A11.** The seasonal average cyclone DpDr for 1985-2014 calculated over 150km x 150km grid boxes for cold and warm seasons.

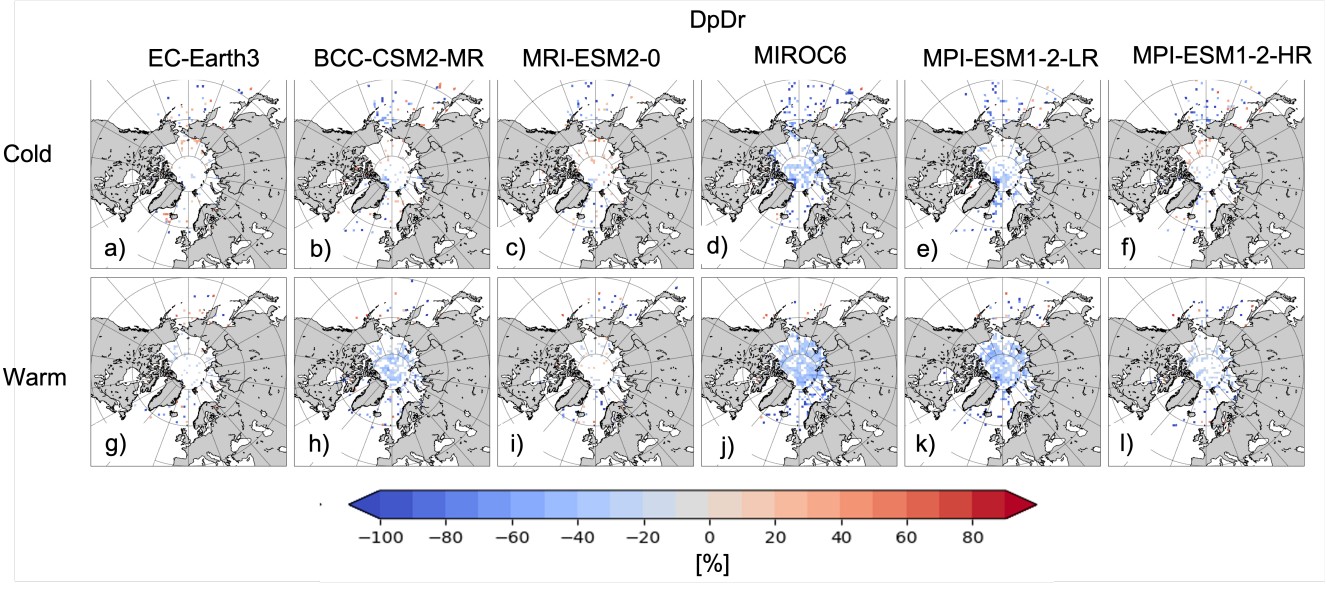

**Figure A12.** The percentage difference between the models and ERA5 for seasonal average DpDr for 1985-2014 calculated over 150km x 150km grid boxes for cold and warm seasons. The statistically significant differences at 90% confidence level are shown.

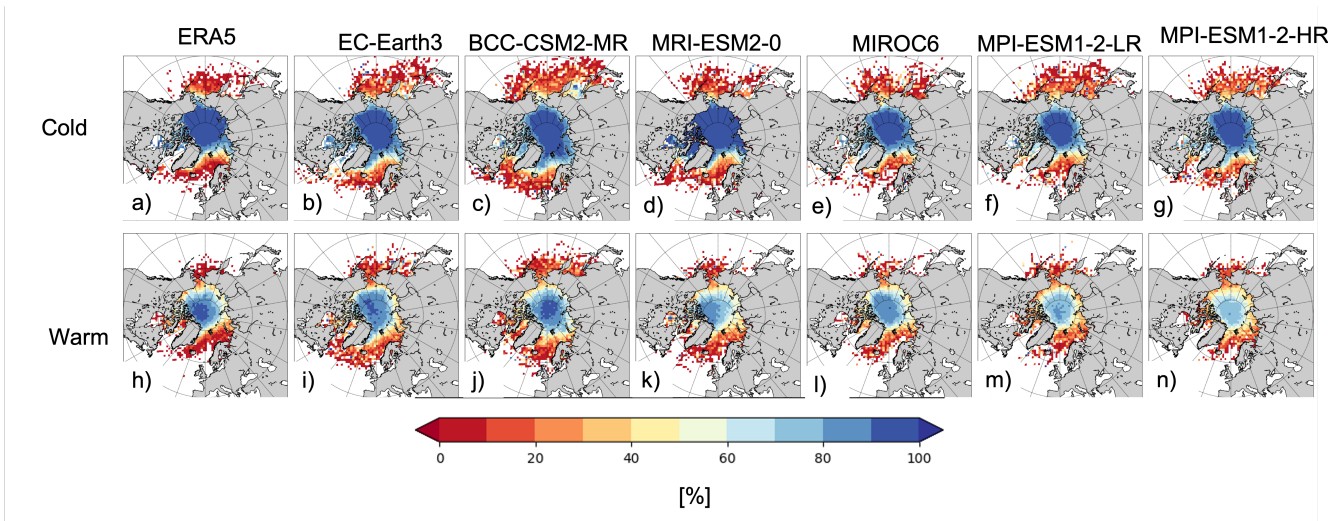

**Figure A13.** The seasonal average SIC calculated over the cyclone area for 1985-2014 calculated over 150km x 150km grid boxes for cold and warm seasons.







**Figure A14.** The lead-lag correlation result.



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
