# Peer review of "Declining Sea Ice and Its Relationship with Arctic Cyclones in Current and Future Climate Part I: Current Climatology in CMIP6 Models"

_Weather and Climate Dynamics, 2023_

## Referee Comment (RC1)

Review of submission entitled "Declining Sea Ice and Its Relationship with Arctic Cyclones in the Current and Future Climate Part I: Current Climatology in CMIP6 Models" by Valkonen et al.

Overall recommendation: Reject

This manuscript aims to assemble a seasonal climatology (1985-2014) from ERA5 reanalysis and output from six coupled CMIP6 models on Arctic cyclone metrics that include cyclone count, minimum central pressure, depth, duration, pressure gradient, cyclone area, and accumulated cyclone energy (or ACE). A cyclone tracker algorithm based mean sea level pressure is applied to each dataset to compute the cyclone metrics, where cyclone area is used to obtain sea-ice concentration (SIC) when the cyclone circulation is co-located with the sea ice. Time series and trends of cyclone metrics and SIC from ERA5 are used to evaluate the same metrics from CMIP6 models. The authors' also refer to a relationship between Arctic cyclones and SIC, which is implicitly established through co-positioning of cyclone area and SIC. While I find the statistical analysis presented on cyclone metrics, ERA5 vs. CMIP6 models, interesting, I do have major concerns with the manuscript in its current form, in particular (1) the described relationship between the cyclone metrics and SIC and (2) the accuracy of the spatial patterns in cyclone counts as shown in Fig. 2. As discussed in the comments below, I believe much of the results presented in the manuscript rest on these concerns and therefore I cannot recommend this study be published in Weather and Climate Dynamics at this time. I do think the authors could improve the manuscript and resubmit after extensive revisions. My overall recommendation is to reject the manuscript at this time.

Major Comments:

1. A relationship between Arctic cyclones and sea ice has been shown by a number of previous studies, many of which are cited by the authors in this manuscript. However, the key details that comprise this relationship and its complexity (e.g., specific cyclone related processes that impose dynamic or thermodynamic impacts on the sea ice) are not well presented in the introduction such that the authors can build upon them with their results and discussion. Analysis of cyclone metrics and their seasonal year-to-year variability against co-located SIC is interesting, but it does not well explain the physical processes between the two. For example, the authors state that cyclone counts and cyclogenesis trends increase from the late 1990s onward in the cold season which is attributed to SIC decline in the warm season. Is the increase in cyclones a function of more available low-level baroclinicity in the autumn-winter months and/or other local processes? No evidence related to the physical drivers is shown. Perhaps the low-level baroclinicity could be computed in cyclone areas that would help explain or corroborate increased cyclone counts and genesis. Perhaps a noncyclone database could be constructed (i.e., see noncyclone methodology described in Finocchio et al. 2020 and Schreiber and Serreze 2020) to further support the statistical relationships. Having SIC stratified by cyclone and noncyclone would give a more robust understanding of whether SIC changes are cyclone related or not.

2. The spatial pattern of ERA5 cyclone counts shown in Fig. 2 a and b seems much different from those shown Valkonen et al. (2021) Fig. b and e (see comparison in pasted graphic below). There is a modest difference in the years used in each climatology, i.e., 1984-2015 shown here versus 1979-2015 shown in Valkonen et al. (2021), but why is there such a discrepancy in the cyclone density patterns? For example, the North Atlantic storm track is a primary cyclone pathway into the Arctic during the cold season (Valkonen et al. 2021; Serreze and Barrett 2008; Zahn et al. 2018, and many others). Here, the cyclone counts are higher in the central Arctic than in the North Atlantic during the cold season? In the warm season, the spatial patterns of cyclone counts look more reasonable with higher counts in the central Arctic, but still not correct. In the warm season, for example, higher cyclone counts should also be found east of Greenland and along south-coast Alaska, but counts are lower in these locations than expected. I could be misunderstanding the color scale or the units, but shouldn't the cyclone density patterns be relatively consistent with Valkonen et al. 2021 and other studies? In addition, these discrepancies in cyclone counts prompt concern as a large portion of the subsequent statistical analysis links to the positioning of cyclone counts shown in Fig. 2.

[Figure]

3. The introduction, methods, and figure captions could benefit from additional and clearer description. For example, cyclone matrix is referenced but never defined. Even referring to Valkonen et al. (2021), I don't find a clear definition of cyclone matrix. I do find a section in Valkonen et al. (2021) paper entitled "cyclone matrix" but no explicit definition. Is it simply the regional boundary in Fig. 1 and cyclones metrics within, including the 24h duration requirement?

Specific comments:

Abstract: No discussion on the cyclone / sea ice relationship in the context of the main results.

Line 30: Perhaps a comma is missing or the following sentence needs to be rephrased.

"They found that the even though all 30 the models did depict a decline in the SIC, the models show a large spread in SIC results, partly due to large internal climate variability and were less consistent with the ERA-Interim results than the SAT"

Line 42: The complex relationship between cyclones and sea ice is not clearly described using previous studies.
"The complex relationship between cyclones and the changing sea ice, and cyclones' important role in the Arctic now and in the future, make it critical to better understand the interactions between Arctic cyclones and sea ice, and how these interactions may change with a warming climate."

Line 77: This sentence needs a citation.
"Studies have also been conducted to better understand how this relationship might change with changes in Arctic climate."

Line 82: "Arctic cyclones and their relationship with Arctic SIC" needs to be unpacked in the introduction.

Line 84: "Arctic cyclone characteristics" would be good to list the characteristics here.

Line 90: Main goal #3. "To assess the CMIP6 models' ability to represent observed relationships between Arctic cyclones and sea ice, and to accurately describe the causalities between the two"
The observed relationships between Arctic cyclones and sea ice are not clearly explained.

Line 116: Section 2.2 "Reanalysis data – ERA5" ERA5 is not a fully coupled model. All six CMIP6 models are fully coupled models. Does this have implications on the results? If so or not, this should be discussed

Line 141: "were cyclone", where cyclones?

Line 146: "cyclone matrix" needs to be defined

Line 147: "recorder", recorded?

Line 148: "SIC over the cyclone area", SIC co-located with the cyclone area?

Line 150: "How intense each cyclone was (weak, normal strong , calculated based on the 25th lowest, interquartile and the top 25th percentile values of ACE over the whole study period), and the average SIC (less than 15%, more than 85%, or in between) were also noted in the cyclone matrix."
Are these metrics used in the analysis or discussed in the results? I don't recall where/how they were used?

Line 158: ACE metric. Is this average surface wind speed? Some additional explanation is needed.

Line 166: Hurrell and Deser (2009) is not in the reference list.

Figures:

The grey shading in the color bar in Figs. 1 and 2 conflicts with grey colored landmass.

Figure 3 shading is not explained. Same with Figs. 5, 6, and 9.

References

Finocchio, P. M., J. D. Doyle, D. P. Stern, and M. G. Fearon, 2020: Short-term Impacts of Arctic Summer Cyclones on Sea Ice Extent in the Marginal Ice Zone. *Geophys. Res. Lett.*, **47**, 1–9, https://doi.org/10.1029/2020GL088338.

Schreiber, E. A. P., and M. C. Serreze, 2020: Impacts of synoptic-scale cyclones on Arctic sea-ice concentration: A systematic analysis. *Ann. Glaciol.*, **61**, 139–153, https://doi.org/10.1017/aog.2020.23.

Serreze, M. C., and A. P. Barrett, 2008: The summer cyclone maximum over the central Arctic Ocean. *J. Clim.*, **21**, 1048–1065, https://doi.org/10.1175/2007JCLI1810.1.

Valkonen, E., J. Cassano, and E. Cassano, 2021: Arctic Cyclones and Their Interactions With the Declining Sea Ice: A Recent Climatology. *J. Geophys. Res. Atmos.*, **126**, 1–35, https://doi.org/10.1029/2020JD034366.

---

## Author Comment (AC1)

Final Response

We want to thank the reviewers for their insightful comments and the suggestions made to improve the manuscript. We have incorporated most of the suggestions made by the reviewers and have added major changes to this response. We have also included responses to all the reviewers' questions below. The reviewer comments/questions are marked in purple and ours in black.

There were a couple of larger themes that were discussed in both of the reviewers' comments, and we want to address those here first, before moving forward with the detailed answers to both reviewers' comments below.

Reviewer comments in violet, authors' responses in black

1: Figure 2 that shows the cyclone counts for all the models and ERA5 for both seasons differs from the similar figure in Valkonen et al. (2021) because it is supposed to. This current figure shows the subset of extratropical cyclones that were used to this study, whereas in the Valkonen et al. (2021) all NH cyclones were depicted. We have updated the figure caption to clearly state this and included a short discussion about how this figure compares to the one in Valkonen et al. (2021) to use as a sort of a validation as suggested by the reviewer 2.

2: Both reviewers suggested more detailed discussion about the different processes related to cyclone-SIC relationship. This was done by adding discussion in the introduction about previous work done in this topic:

*"The complexity of the cyclone-sea-ice relationship is related to the multiple different processes through which the two can interact. The cyclones passing the sea ice forces the surface, leading to increased or decreased SIC depending on the season, location of the ice with respect to the cyclone location, the age of the ice, or the form of the precipitation associated with the cyclone, for example (Lukovich et al., 2021; Schreiber and Serreze, 2020; Webster et al., 2019). The processes responsible for increasing or decreasing SIC can be thermodynamic, taking place through changes in the radiative balance of the surface due to increased cloud coverage, moisture influx, or changes in temperatures associated with the cyclone (Boisvert et al., 2016; Blanchard-Wrigglesworth et al., 2022); or dynamic through the convergent or divergent movement of the ice due to the winds and Ekman transport associated with them (Blanchard-Wrigglesworth et al., 2022 ). Which process ends up being the dominant one, appears to depend on the observed timescale. For example, Schreiber and Serreze. (2020) studied the effects synoptic-scale cyclones had on Arctic sea ice on 4-day time scales (averaged seasonally) and found that in the summer season cyclones decreased sea ice melt due to thermodynamic effects, while Finnochio et al. (2022) studied May-August cyclones with 1-5 day influence and found that thermodynamic effect was decelerating ice decline in May-June, while dynamic effects became important in July and August. In addition, changing ice influences the atmosphere and can therefore affect the passing cyclones (Koyama et al., 2017). These atmospheric changes include surface temperature, cloud cover, and radiative balances changes, increases in atmospheric moisture content, decreases in static stability, and changing turbulent fluxes (Schweiger et al., 2008; Rinke et al., 2006; Koyama et al., 2017, Messori et al., 2018). In addition to affecting moisture transport and surface temperatures and energy fluxes in the Arctic (Messori et al.,*

*2918), these variations can influence the passing cyclones through changes in baroclinicity, surface roughness, or atmospheric moisture content for example (). An opening in the ice layer increases the surface turbulent fluxes, decreases static stability, and causes changes in surface radiation. This can lead to an increase in near-surface baroclinicity and a more suitable environment for cyclone development (Rinke et al., 2017), but also faster ice growth, as new ice grows faster than multiyear ice or decrease in ice melt due to radiative effects (Schreiber and Serreze (2020). These two-way interactions could also be altered by strong Arctic amplification and thinning ice pack (Parker et al., 2023).”*

We also made sure to more clearly describe the statistical trend and correlation analysis described in section 3.3 and added a new figure (Fig. 11, shown below) that showed a summary of the correlation analysis to make the discussion about the main findings clearer. References to previous work and how it relates to findings in section 3.3 (and other result sections) were also added.

[Figure]

3: The explanation of the methods was made clearer by removing any references to the 'cyclonematrix', writing out the requirements for Arctic cyclones more clearly and defining the ACE more precisely as follows. These changes are written out below in the detailed responses to each reviewer.

Reviewer 1:

Review of submission entitled "Declining Sea Ice and Its Relationship with Arctic Cyclones in the Current and Future Climate Part I: Current Climatology in CMIP6 Models" by Valkonen et al.

Overall recommendation: Reject
This manuscript aims to assemble a seasonal climatology (1985-2014) from ERA5 reanalysis and output from six coupled CMIP6 models on Arctic cyclone metrics that include cyclone count, minimum central pressure, depth, duration, pressure gradient, cyclone area, and accumulated cyclone energy (or ACE). A cyclone tracker algorithm based mean sea level pressure is applied to each dataset to compute the cyclone metrics, where cyclone area is used to

obtain sea-ice concentration (SIC) when the cyclone circulation is co-located with the sea ice. Time series and trends of cyclone metrics and SIC from ERA5 are used to evaluate the same metrics from CMIP6 models. The authors' also refer to a relationship between Arctic cyclones and SIC, which is implicitly established through co-positioning of cyclone area and SIC. While I find the statistical analysis presented on cyclone metrics, ERA5 vs. CMIP6 models, interesting, I do have major concerns with the manuscript in its current form, in particular (1) the described relationship between the cyclone metrics and SIC and (2) the accuracy of the spatial patterns in cyclone counts as shown in Fig. 2. As discussed in the comments below, I believe much of the results presented in the manuscript rest on these concerns and therefore I cannot recommend this study be published in Weather and Climate Dynamics at this time. I do think the authors could improve the manuscript and resubmit after extensive revisions. My overall recommendation is to reject the manuscript at this time.

Major Comments:

1. A relationship between Arctic cyclones and sea ice has been shown by a number of previous studies, many of which are cited by the authors in this manuscript. However, the key details that comprise this relationship and its complexity (e.g., specific cyclone related processes that impose dynamic or thermodynamic impacts on the sea ice) are not well presented in the introduction such that the authors can build upon them with their results and discussion.

We have increased the discussion of thermodynamical vs dynamical processes and provided more detail discussion about the complexity of the cyclone sea ice relationship in the introduction as shown above. We have also included some of this discussion in the results section, where appropriate.

Analysis of cyclone metrics and their seasonal year-to-year variability against co-located SIC is interesting, but it does not well explain the physical processes between the two. For example, the authors state that cyclone counts and cyclogenesis trends increase from the late 1990s onward in the cold season which is attributed to SIC decline in the warm season. Is the increase in cyclones a function of more available low-level baroclinicity in the autumn-winter months and/or other local processes? No evidence related to the physical drivers is shown. Perhaps the low-level baroclinicity could be computed in cyclone areas that would help explain or corroborate increased cyclone counts and genesis.

The reviewers bring up an important point about the physical processes between the SIC and cyclones. However, as described in the introduction section of the paper, the main goals of this specific paper are to describe the cyclone and co-located SIC climatologies and how they are represented in the models, not necessarily to provide detail description of the physics behind the cyclone-SIC interaction. In order to study the physics in the models, one needs first understand how these climatologies are presented in them, which is the main goal of this paper. Future work

could then be done to quantify the physical drivers between the cyclones and sea ice. We have added discussion on possible physical mechanisms for the observed cyclone-SIC relationship based on previous work and discussed how it relates to the results shown in this manuscript. The discussion about correlation results has been changed to:

*"Lead-lag correlation analysis was also performed between cyclone counts, SIC and NAO in order to assess if differences in large-scale circulation, as represented by the NAO, are related to the cyclone count errors. Figure 11 shows the results for ERA5 and multi-model mean. The full results are shown in the appendices (Fig. A14). The lead-lag analysis showed strong correlation between cyclone counts and SIC in the ERA5 data, with weaker relationship between the cyclone counts and NAO. The cyclone count-SIC correlations in ERA5 were stronger when lagged, suggesting a positive feedback loop between the increased cyclone counts and SIC, where less sea ice is related to higher cold season cyclone counts and higher cyclone counts in cold season are associated with reduced SIC in both seasons. This is consistent with the findings by Valkonen et al. (2021), who found similar lagged correlations between cold season cyclones and SIC in three different reanalysis products between 1979-2015. The negative correlations, which are stronger with seasonal lags, could be happening because more cyclones in the cold season are enhancing ice reduction through thermodynamic and dynamic forcings (Lukovich et al., 2021). As mentioned, individual cyclones can cause breaks in the ice, inducing higher turbulent heat fluxes from the open ocean, changing surface radiative balance, and altering ocean circulation through the surface winds leading to higher seasonal SIC decline (Blanchard-Wrigglesworth et al., 2013). The strong decrease of sea ice in the cold season and related atmospheric changes can then affect the melt season in the following spring and summer (Holland and Stroeve, 2011; Mortin et al., 2016), delaying the start of the freeze-up in the following fall. This could be the reason for the high correlation between increased cyclone counts in the cold season and lower SIC in the following warm season. As found by Koyama et al. (2017) the lower SIC in the summer can then increase baroclinicity in the following winter, which could fuel local cyclogenesis. This positive feedback loop could explain the lead-lag correlations, trendmatrices and the strong increase of Arctic cyclogenesis in the cold season from 2000s onward observed in this study.*
*In contrast for the CMIP6 models, SIC was not significantly correlated with the cyclone counts in any season or lag. This means that there is no positive feedback loop relating SIC decline and cyclone counts to each other. This is similar to what is observed in the trendmatrices (Fig. \ref{fig10}) and signifies that the CMIP6 models are not reproducing the relationship between Arctic cyclones and SIC accurately. Given that we observed the models underestimating Arctic cyclogenesis; together with larger underestimations during times that sea ice has been melting (the warm season; in the cold season after the early 2000s), we hypothesize that the CMIP6 models are not fully able to present processes that relate sea ice melt to cyclone formation in the Arctic. The processes could include parametrization of turbulent fluxes or surface drag, but also more regional processes, such as lee-cyclogenesis from Greenland could be lacking."*

> Perhaps a noncyclone database could be constructed (i.e., see noncyclone methodology described in Finocchio et al. 2020 and Schreiber and Serreze 2020) to further support the statistical relationships. Having SIC stratified by cyclone and noncyclone would give a more robust understanding of whether SIC changes are cyclone related or not.

The non-cyclone database is a good idea and can definitely help with more detailed understanding of the influence a cyclone has on the sea ice in certain cases. However, in this paper we wanted to focus on the two-way relationship between cyclones and sea ice, not just the top-down effect, which is why full climatologies were included. In addition, since we are

calculating the lead/lag correlations for full datasets, the times without cyclone present are also included in this analysis. We have added more discussion on the correlation analysis with a focus on the statistical relationships between the cyclones and SIC as shown above.

> 2. The spatial pattern of ERA5 cyclone counts shown in Fig. 2 a and b seems much different from those shown Valkonen et al. (2021) Fig. b and e (see comparison in pasted graphic below). There is a modest difference in the years used in each climatology, i.e., 1984-2015 shown here versus 1979-2015 shown in Valkonen et al. (2021), but why is there such a discrepancy in the cyclone density patterns? For example, the North Atlantic storm track is a primary cyclone pathway into the Arctic during the cold season (Valkonen et al. 2021; Serreze and Barrett 2008; Zahn et al. 2018, and many others). Here, the cyclone counts are higher in the central Arctic than in the North Atlantic during the cold season? In the warm season, the spatial patterns of cyclone counts look more reasonable with higher counts in the central Arctic, but still not correct. In the warm season, for example, higher cyclone counts should also be found east of Greenland and along south-coast Alaska, but counts are lower in these locations than expected. I could be misunderstanding the color scale or the units, but shouldn't the cyclone density patterns be relatively consistent with Valkonen et al. 2021 and other studies? In addition, these discrepancies in cyclone counts prompt concern as a large portion of the subsequent statistical analysis links to the positioning of cyclone counts shown in Fig. 2.

It is true that Figure 2 in this paper does not match with FIG 1b in Valkonen et al. (2021). This is because the datasets used in these two papers are inherently different. For the 2021 paper in the spatial plots, we showed all North Atlantic cyclones that were detected by the tracker to show the spatial patterns and comparisons between the 3 reanalysis. In this paper, however, we wanted to only focus on the cyclones that were used for the other analyses; the so called "Arctic Cyclones", those that existed for 24 hours or more inside the yellow area shown in figure 1. So, basically the figure 2 in this paper is a subset of tracks shown in Figure 1b in the 2021 paper. This is also why the figure differs from any other papers as many other papers define Arctic as North of 60N, which will allow for more cyclones to be tracked.  We will explain this better in the text and update the figure captions also.

> 3. The introduction, methods, and figure captions could benefit from additional and clearer description. For example, cyclone matrix is referenced but never defined. Even referring to Valkonen et al. (2021), I don't find a clear definition of cyclone matrix. I do find a section in Valkonen et al. (2021) paper entitled "cyclone matrix" but no explicit definition. Is it simply the regional boundary in Fig. 1 and cyclones metrics within, including the 24h duration requirement?

Thank you for the comment. "Cyclone matrix" means the dataset of cyclones that fulfil the 24h within the Arctic (the area described in FIG 1) criteria, and the metrics calculated for each of those cyclones. Since this phrase is a bit confusing and does not really bring anything more to the paper, we have removed it from the paper, and have also update the descriptions to be more comprehensive and easily understandable. The updated definitions are as following:

"Following Valkonen et al. (2021) the cyclone tracking results are post-processed to gain a better understanding of the Arctic cyclones. The tracking algorithm runs globally, but the focus of this study is the Arctic region. The study area is shown by a yellow line in Figure \ref{fig1}. Only cyclones that exist for 24 hours or more in this study region are included in the following analysis. For these cyclones multiple characteristics, such as time, location, lifetime and cyclone area are recorded for every time-step over the cyclone's lifetime. Cyclones are then divided into cold season (December - May) and warm season (June-November) events based on the month they existed in. In addition, average SIC calculated over the cyclone area is recorded.

To assess cyclone intensities multiple metrics are also kept…"

> Specific comments:
> Abstract: No discussion on the cyclone / sea ice relationship in the context of the main results.

We have added some discussion about the cyclone-SIC relationship in the abstract:
*"…co-existent Arctic cyclone characteristics that were evident in the reanalysis data. The local cyclogenesis in the Arctic was shown to be underestimated, which led to an overall underestimation of Arctic cyclones in the CMIP6 model results. We hypothesized that this could be due to the models lack of properly depict the coupled relationship between the declining sea ice and increasing cold season cyclones. Increased cyclogenesis in the cold season was found to be related to less sea ice in the following warm season, which could then affect cyclogenesis further on."*

> Line 30: Perhaps a comma is missing or the following sentence needs to be rephrased. "They found that the even though all 30 the models did depict a decline in the SIC, the models show a large spread in SIC results, partly due to large internal climate variability and were less consistent with the ERA-Interim results than the SAT"

We have rephrased the sentence as follows: "They found that all 30 models did depict a decline in the SIC. However, the models showed a large spread in SIC results, partly due to large internal climate variability, and SIC results were less consistent with the ERA-Interim results than the SAT result were."

> Line 42: The complex relationship between cyclones and sea ice is not clearly described using previous studies. "The complex relationship between cyclones and the changing sea ice, and cyclones' important role in the Arctic now and in the future, make it critical to better understand the interactions between Arctic cyclones and sea ice, and how these interactions may change with a warming climate."

We have added more discussion about the relationships between SIC and cyclones emphasizing previous works as shown above. We also paid more attention to include both 'cyclone to sea ice' and 'sea ice to cyclone' -interactions to emphasize the complexity of this relationship.

> Line 77: This sentence needs a citation. "Studies have also been conducted to better understand how this relationship might change with changes in Arctic climate."

Added citations, sentence now reads : "Studies have also been conducted to better understand how this relationship might change with changes in Arctic climate (Parker et al., 2022; Valkonen et al., 2021; Lukovich et al., 2021; Screiber and Serreze, 2020) ."

> Line 82: "Arctic cyclones and their relationship with Arctic SIC" needs to be unpacked in the introduction.

We have added discussion about this as stated in the beginning.

> Line 84: "Arctic cyclone characteristics" would be good to list the characteristics here.

Added the characteristics: "Arctic cyclone characteristics: counts, genesis counts, intensities, and radii;"

> Line 90: Main goal #3. "To assess the CMIP6 models' ability to represent observed relationships between Arctic cyclones and sea ice, and to accurately describe the causalities between the two" The observed relationships between Arctic cyclones and sea ice are not clearly explained.

We have rephrased the main goal #3 to be more in line with the content and purpose of the paper as follows: "To assess the CMIP6 models' ability to represent observed relations between Arctic cyclones and sea ice."

> Line 116: Section 2.2 "Reanalysis data – ERA5" ERA5 is not a fully coupled model. All six CMIP6 models are fully coupled models. Does this have implications on the results? If so or not, this should be discussed

It is true that ERA5 is not a fully coupled model, but as observations are assimilated into the model it gives a good understanding of the "true" state of the atmosphere. ERA5 model was also compared to CFSR model in Valkonen et al. (2021), which is a fully coupled model and it was shown that while ERA5 shows some differences in general results are comparable. This was added to the 2.2. section.

> Line 141: "were cyclone", where cyclones?

Yes, corrected the typo.

> Line 146: "cyclone matrix" needs to be defined

We have removed all references to the cyclone matrix as described above. We rephrased the ending of that sentence to "…were included."

> Line 147: "recorder", recorded?

Typo corrected.

Line 148: "SIC over the cyclone area", SIC co-located with the cyclone area?

Yes, we have rephrased the sentence as following: "average SIC calculated over the cyclone area…"

Line 150: "How intense each cyclone was (weak, normal strong , calculated based on the 25th lowest, interquartile and the top 25th percentile values of ACE over the whole study period), and the average SIC (less than 15%, more than 85%, or in between) were also noted in the cyclone matrix." Are these metrics used in the analysis or discussed in the results? I don't recall where/how they were used?

This description was removed as they are not pertinent to the analysis here.

Line 158: ACE metric. Is this average surface wind speed? Some additional explanation is needed.

We have updated the ACE definition as following:
"The ACE is defined following Klotzbach (2006), where the maximum wind speed of a cyclone at each time step during the cyclone lifetime is squared. ACE is therefore proportional to the kinetic energy (per unit mass) of the wind field.  In this study we calculate the ACE-metric based on the mean squared wind speeds over the cyclone area instead of the cyclone maximum wind speed as shown by Eq. 1.

$$ACE = mean((V_{i,j})^2) \qquad\qquad (1)$$

In Equation 1 $V_{i,j}$ describes the wind speed at each grid point within the cyclone area."

Line 166: Hurrell and Deser (2009) is not in the reference list.

Reference was added to the reference list.

Figures: The grey shading in the color bar in Figs. 1 and 2 conflicts with grey colored landmass.

We are not certain what is meant by the grey shading in color bar, as there is no grey shading in these figures.

Figure 3 shading is not explained. Same with Figs. 5, 6, and 9.

We have added the explanation of the blue shading on those figures: "… The blue shading depicts the 95th quartile spread of the multi-model mean."

Reviewer 2:

**Review notes for the manuscript to WCD: "Declining sea ice and its relationship with Arctic cyclones in current and future climate Part 1: Current climatology in CMIP6 models" by Valkonen et al.**

**General comments**

This paper presents an Arctic cyclone and sea ice climatology for 1985 – 2014 utilizing a chosen ensemble of CMIP6 simulations. The authors further use ERA5 reanalysis data to address how closely the model succeeds in resembling characteristics of and relationships between sea ice and Arctic cyclones. The authors find that the sea ice trend is well reproduced in the models, whereas the coupling between sea ice and Arctic cyclones are less represented. Specifically, the authors find the models to struggle with representing local cyclogenesis and cyclone intensities; the former leading to an underestimation of Arctic cyclones compared to ERA5. This study tries to complete existing studies where the relationship between Arctic cyclones and sea ice is addressed, both in models and in observational data, in the current and in a changing "new" Arctic. However, in a main attempt to use CMIP6 to investigate model performance of Arctic cyclone characteristics and the relationship between cyclones and sea ice (which has been partly touched upon in previous studies as the authors point out in the introduction) and further trying to discuss causalities, in my opinion the authors fail to address this properly. This concerns me a little. The main reason to this is that the authors focus on the impact of sea ice on cyclogenesis, but never discuss the two main processes Arctic cyclones impose *on* the sea ice (see specific comments below; processes that can have different effect on the sea ice depending on the time scale and the local region). Thus, quantifying biases between models and ERA5 and addressing the causalities for the relationship without quantifying the two main processes lead to that a large (important) part of this paper is missing, in my opinion. The authors discuss biases they find mostly wrt the intensity metrics' and how for example the biases in the depth of the cyclones (models overestimate) are related to the location of local cyclogenesis (underestimation of local cyclogenesis) – this is interesting and new findings, but need to be complemented with additional analysis. Additionally, even though the figures are well done, they do not always support the main findings of this paper. Therefore, I suggest a major revision before any possible acceptation of the paper. See comments below for clarifications.

**Specific comments**

- As the title "declining sea ice and its relationship with Arctic cyclones" already suggest a link between SIC and frequency of Arctic cyclones, me as a reader would expect this link to be more discussed in the paper. The authors multiple times mention the "one way" relationship, i.e., a potential for more local cyclogenesis and higher cyclone frequency at the marginal ice zones due more baroclinic zones following the declining sea ice (how the declining sea ice affects local cyclogenesis: discussed briefly at L240). The authors focus on the time scale between years (if I understand correctly) and claim that an increase in the cyclone frequencies is due to the declining sea ice trend (where the cyclone characteristics not well captured by models). However, there is also a two-way relationship between SIC and cyclones which the authors do not mention at all. This is, Arctic cyclones also give rise to local changes on the sea ice, which depends on the timescale, local location, cyclone intensity and local sea ice conditions. There is an attempt in addressing the role of cyclones for the Arctic climate and the relationship to sea ice changes in the paragraph starting at L32, but the main processes cyclones impose on the sea ice are missing. These two processes – thermodynamical (ice phase changes and relation to surface energy fluxes) and dynamical (wind induced) processes – show different effects on sea ice changes depending on cyclone tracks and the time scales (shorter, immediate impact or with longer timescales of weeks after a cyclone has passed). I suggest the authors to read e.g., https://doi.org/10.1029/2022GL100051 from Aue et al. 2022. There, the overall impact of cyclones on the sea ice in the Atlantic sector is discussed. The main findings are that dynamical processes dominate at the direct impact of the cyclone (sea ice loss; wind causing sea ice openings, but also thermodynamical sea ice loss in the warm sector of the cyclone), whereas thermodynamical processes dominate weeks after the cyclone has passed. These "post-" processes are mainly in the cold sector, with *positive* sea ice changes as the sea ice refreezes in the openings of the Barents Sea.

We have added discussion about the two-way processes that are related to the sea ice cyclone relationship both in the introduction (as shown at the beginning of the response) and to the discussion sections. The trend and correlation analyses were done to depict/analyze the two-way interactions between cyclones and sea ice, and their relevance and importance is discussed more in detail as shown below:

*"Lead-lag correlation analysis was also performed between cyclone counts, SIC and NAO in order to assess if differences in large-scale circulation, as represented by the NAO, are related to the cyclone count errors. Figure 11 shows the results for ERA5 and multi-model mean. The full*

*results are shown in the appendices (Fig. A14). The lead-lag analysis showed strong correlation between cyclone counts and SIC in the ERA5 data, with weaker relationship between the cyclone counts and NAO. The cyclone count-SIC correlations in ERA5 were stronger when lagged, suggesting a positive feedback loop between the increased cyclone counts and SIC, where less sea ice is related to higher cold season cyclone counts and higher cyclone counts in cold season are associated with reduced SIC in both seasons. This is consistent with the findings by Valkonen et al. (2021), who found similar lagged correlations between cold season cyclones and SIC in three different reanalysis products between 1979-2015. The negative correlations, which are stronger with seasonal lags, could be happening because more cyclones in the cold season are enhancing ice reduction through thermodynamic and dynamic forcings (Lukovich et al., 2021). As mentioned, individual cyclones can cause breaks in the ice, inducing higher turbulent heat fluxes from the open ocean, changing surface radiative balance, and altering ocean circulation through the surface winds leading to higher seasonal SIC decline (Blanchard-Wrigglesworth et al., 2013). The strong decrease of sea ice in the cold season and related atmospheric changes can then affect the melt season in the following spring and summer (Holland and Stroeve, 2011; Mortin et al., 2016), delaying the start of the freeze-up in the following fall. This could be the reason for the high correlation between increased cyclone counts in the cold season and lower SIC in the following warm season. As found by Koyama et al. (2017) the lower SIC in the summer can then increase baroclinicity in the following winter, which could fuel local cyclogenesis. This positive feedback loop could explain the lead-lag correlations, trendmatrices and the strong increase of Arctic cyclogenesis in the cold season from 2000s onward observed in this study.*

*In contrast for the CMIP6 models, SIC was not significantly correlated with the cyclone counts in any season or lag. This means that there is no positive feedback loop relating SIC decline and cyclone counts to each other. This is similar to what is observed in the trendmatrices (Fig. 10) and signifies that the CMIP6 models are not reproducing the relationship between Arctic cyclones and SIC accurately. Given that we observed the models underestimating Arctic cyclogenesis; together with larger underestimations during times that sea ice has been melting (the warm season; in the cold season after the early 2000s), we hypothesize that the CMIP6 models are not fully able to present processes that relate sea ice melt to cyclone formation in the Arctic. The processes could include parametrization of turbulent fluxes or surface drag, but also more regional processes, such as lee-cyclogenesis from Greenland could be lacking."*

> Also, the abstract of this current manuscript is missing the most important links to the processes that cyclones impose on sea ice (when talking about relationships: "The model results did closely match reanalysis data in depicting the observed sea ice trend. However, we found that the model results struggled to reproduce the strongly coupled relationship between the declining sea ice and Arctic cyclones"). This is all fine, but proof to the second sentence is partly missing in the paper. Discussing the relationships between SIC and cyclones need to include both-way links. Thus, when these two important processes are not quantified (or even discussed in the paper), causalities are hard to address

(if only thinking about the role of sea ice loss on local cyclogenesis). I suggest the authors to consider the two-way relationship and add more analysis to the current manuscript.

Thank you for the comments, we have added discussion about the two-way interactions to the paper both in the introduction and where relevant in the result sections. We have also emphasized the trendmatrix and correlation results (Figure 10 and new Figure 11) more, to underline that in the reanalysis data there is a statistical relationship between the two that is not seen in the CMIP6 model data.

- The authors also compare the negative trends of SIC and the relation with cyclone frequencies. I would be happy to see more discussion about how does the changed "new Arctic" with thinner and lower sea ice concentration affect the impact of cyclones compared to the "old Arctic"? Again, a two-way relationship (SIC --> cyclones and cyclones --> SIC changes) is suggested here.

As shown above we have added discussion about the two-way relationship between cyclones and SIC and the difference in cold season cyclone counts after early 2000s ("new Arctic").

- In the introduction, I am also missing the discussion between cyclones, moisture transport and extreme surface temperatures (as discussed in e.g., in the context of drivers for warm spells in Messori et al. 2018: https://doi.org/10.1175/JCLI-D-17-0386.1), as well as present some case studies of how cyclones affect the sea ice e.g., Boisvert et al. 2016 (showing the potential of a local thermodynamical ice loss due to anomalous energy flux towards the surface, but also sea-ice retreat induced by dynamical forcing by the cyclone winds contribute to the observed sea ice loss: https://doi.org/10.1175/MWR-D-16-0234.1). I find some attempt in the manuscript to highlight the different findings between the occurrence of Arctic cyclones and sea ice changes from L69 and a case study at L73 (emphasizing the increase in SIC after the cyclone passed), but no discussion of the reasons behind or the two main processes cyclones impose on the sea ice and/or the temporal time scales considered. I suggest the authors to complete the introduction as well as the discussion parts with the missing parts.

As mentioned, we added discussion about the different interactions between cyclones and sea ice in the introduction section (as shown at the beginning of this response), and where relevant in the results and discussion sections.

- The authors quantitatively describe biases between models and ERA5. I was wondering if the authors know any studies where the representativity or limitation of ERA5 in depicting cyclones is shown?

There have been studies addressing different aspects of the ERA5 reanalysis data (Herrmannsdöfer et al., 2023; Campos et al., 2022 for example). Rohrer et al. (2019) looked at synoptic scale blocks and cyclones in ERA5 and found that ERA5 is consistent with ERA-Interim data. Valkonen et al., 2021 also compared ERA5 to two other reanalysis product finding good correspondence between the three. However, we are not aware of studies utilizing direct observational or satellite data at this point.

- An interesting finding of this current paper is the differences they find in the model intensities compared to ERA5. I was firstly wondering if the authors could quantify which of the intensity metrices they describe are more important? If the cyclone frequencies are underestimated by models but the central pressure is also underestimated together with an overestimation of the radius (models find stronger and deeper cyclones with larger area); which one of the intensities metrices are more important (depth, frequency, area, or energy flux?). In the process of quantifying the biases, I would like to see a discussion where relative importance of each metrics is discussed.

The reviewer makes an important point about the relative importance of each intensity metric and we have added discussion about the relevance of each metric to the results section. As for to quantify, which metric is the most important, we don't think that would be possible to do. The importance of each metric is dependent on the focus of individual studies/what is the use purpose for the cyclone dataset. Due to this open nature of the question we decided to show all these different metrics, instead of picking one metric to study.

> Secondly, the authors elaborate on the reasons behind the differences between metrices (e.g., deeper and larger cyclones in models), and emphasise mainly on the representation of the SLP field and the resolution of the models. I find these results quite interesting. What do the authors think about other reasons behind the differences? For example, how well are small-scale processes such as diabatic heating represented by the models (which in a warming climate could lead to a potential of a larger moisture content and thus strengthening of cyclones through PV production via diabatic heating might be even more important in the future?) See e.g., Dominic Bühler, Stephan Pfahl (2017) https://doi.org/10.1175/JAS-D-17-0041.1 about extratropical cyclones that explains processes linked to PV diagnostics and cyclone intensification. Of course, these are mainly extratropical cyclones and the current manuscript discusses Arctic cyclones – but maybe still something to consider also here?

Different small-scale processes could very well be influencing the deepening of the cyclones and partly explain the biases that are seen in the SLP, central pressure and depth metrics (Table 1). However, when we study Figure 8, we see that most of the under/overestimations of SLP appear to be consistent over large swaths of the Arctic, and strongly connected to the cyclone central pressure biases observed in the models. This hints us that the SLP biases in the models are systemic level issues and are hence beyond the scope of this study. We will add more discussion on what issues have been detected so far on the different CMIP6 models so far. The fact that for certain metrics (especially depth) their genesis location (mid-latitude/Arctic) plays a role on their intensity does hint that there might be mid-latitude processes that differ from Arctic and influence the cyclone deepening.

> Also, other processes, such as winds from Greenland over Arctic sea ice could lead to local cyclone genesis- again processes that might not be that well captured/parameterized in models and thus cyclones are underestimated in models? I suggest authors to elaborate a bit more about the reasons behind the differences.

The underestimation of local genesis in the Arctic appeared to be mostly evenly distributed over the whole Arctic (Fig. 4 a compared to c-h and b compared to i-n) and larger in cases with higher temperatures/lower SIC. This implies that processes that are responsible/badly represented in the models must be taking place throughout the Arctic and be related to changing temperatures/sea ice. However, regional processes could definitely

- Cyclone postprocessing method is for me a bit unclear (L144): the authors say that it is done following a previous study. However, not having read that paper, it is unclear for me if the postprocessing includes the sub-selection of cyclones based on their duration and spatial location or what it may include. Please rewrite these sentences for clarification.

We have rewritten this part as: "Following Valkonen et al. (2021) the cyclone tracking results are post-processed to gain a better understanding of the Arctic cyclones. The tracking algorithm runs globally, but the focus of this study is the Arctic region. The study area is shown by a yellow line in Figure 1. Only cyclones that exist for 24 hours or more in this study region are included in the following analysis. For these cyclones multiple characteristics, such as time, location, lifetime and cyclone area are recorded

for every time-step over the cyclone's lifetime. Cyclones are then divided into cold season (December - May) and warm season (June-November) events based on the month they existed in. In addition, average SIC calculated over the cyclone area is recorded.

To assess cyclone intensities multiple metrics are also kept"

- Sometimes the authors are mentioning twice the method of a certain parameter, e.g., the cyclone intensities on L150 and L153 – please modify and rephrase so that the message comes through. For example, rewrite on L153: "additionally to the ACE, other metrices for determining cyclone intensities were used. These metrices include cyclone central pressure, …" Also, the end of the paragraph needs to be re-written or moved to the begin of the paragraph (where the ACE is discussed).

Done, these sentences were removed, as they more related to the cyclone matrix discussion.

- Regarding the ACE, for me it is a bit unclear what kind of intensity metrics it actually is. To my understanding from the manuscript, the ACE tells about the transfer of momentum and heat between the surface and the cyclone. However, it is unclear *how* it is calculated from the mean squared wind speeds (as given in L159). Please clarify.

We have rephrase the ACE definition as follows: "To capture this relationship more robustly we use the Accumulated Cyclone Energy (ACE) metric as an additional intensity metric for the cyclones. The ACE is defined following Klotzbach (2006), where the maximum wind speed of a cyclone at each time step during the cyclone lifetime is squared. ACE is therefore proportional to the kinetic energy (per unit mass) of the wind field.  In this study we calculate the ACE-metric based on the mean squared wind speeds over the cyclone area instead of the cyclone maximum wind speed as shown by Eq. 1.     $\mathrm{ACE} = \mathrm{mean}((V_{i,j})^2)$ (1). In Equation 1 $V_{i,j}$ describes the wind speed at each grid point within the cyclone area."

- I was wondering why the NAO index was calculated by the authors and not using the daily NAO index provided by NOAA (https://www.cpc.ncep.noaa.gov/products/precip/CWlink/daily_ao_index/teleconnections.shtml)?

NAO was used in the correlation analysis (Fig. A14), so it was needed for each model separately, something that the NOAA index couldn't provide.

- Just a clarification question: are the cyclone counts provided in Fig 2 shown for the (lon,lat) points of the cyclone tracks identified by the minimum SLP, whereas the cyclone area considers a larger area (defined by the last closed isobar)?

Yes, exactly so.

- Suggestion: it would be nice to see cyclone frequencies over the whole northern hemisphere for the two seasons (for reference to the reader) – not only those selected to be within the Arctic region (Fig 2). Compare Fig. 4 in Wernli and Schwierz 2006 (Journal of the Atmospheric Sciences). However, the frequencies shown in Fig. 2 of this paper in review shows similar frequencies for the Arctic as their Fig. 4. If the authors decide not to include an additional panel in Fig2, I suggest that the other figure is discussed for comparison and "validation" of the authors cyclone data.

Thank you for the suggestion. This is something that was done for the ERA5 (and ERA-Interim, CFSR) reanalysis for the Valkonen et al (2021) paper. We have included a short discussion with a reference to that paper.

- For the readers convenience and to support the main findings of the paper (underestimation of cyclones in models compared to ERA5), I would suggest adding the difference plot in cyclone frequencies (ERA5 minus in models) for the two seasons already in the main paper. Suggest that one representative CMIP6 model is chosen.

Thank you for the good suggestion. This has been done for all the models and this figure can be found in the appendices (Fig. A1).

- In section starting on L380, the relationships between SIC changes and cyclone frequencies are stated, however, no mention of correlation analysis or similar mathematical methods for defining this relationship are presented. Also, I am confused what the "time range" is referring to on the y-axis of Fig. 10. As the impact on cyclones on the SIC changes depending on the time scale considered (on lag times after a cyclone has passed), I suggest the authors to add this information as well.

In the section in question (3.3) the cyclone-SIC relationships are discussed with respect to the trend calculations (Fig. 10) over multiple different time scales and start years (to guarantee the robustness of any trends observed) and lead-lag correlation analysis (Fig. A14). In order to make this more clear we have updated the description of Figure 10 to:

"Trend matrices for cyclone counts and SIC. Panels a,c-h show statistically significant trends for cyclone counts and panels b, i-n for SIC. In each panel the x-axis displays the start year of the trend calculation and y-axis over how many years the calculation is done for."

and added a new Figure 11 that will show the results of Figure A14 for ERA5 and multi-model mean for cyclone counts (Fig A14 counts).

**Technical corrections**

- E.g., at L15: "high North": wondering if this is an appropriate notation. I assume the authors aim to say "the high Arctic"?

Changed "high North" to "Arctic"

- Unclear what "describe the causalities between the two" in Point (3) at L91 refer to – to the relationship between Arctic cyclones and sea ice, or why we see the differences between CMIP6 models and ERA5?

This sentence was removed.

- The reason for the selection of the CMIP6 models are presented in two separate sentences (L100, L105) – not sure which one is more dominant. Please rewrite / write more concise.

Rephrased sentences in L100-111 to "In this study six models were chosen from the CMIP6 Project and are fully couple global models with interactive oceans and sea ice."

- The section about the "history" of ERA5 (from L117) seems a bit too much, in my opinion. If the authors wish to include this in the paper, maybe consider to mention (some) of it in the introduction. Here, I would just mention the data, resolution of the data as well as discuss its representativity and/or possible limitations.

The discussion from L117-121 was removed. Section 2.2 now states: "In this study the ensemble of CMIP6 models were compared against the ERA5 reanalysis product (Hersbach et al., 2020), which is the newest reanalysis from the European Centre for Medium-Range Weather Forecasts (ECMWF) and is the highest global resolution reanalysis product to date. ERA5 has a horizontal resolution of 31 km (T639) and 137 vertical levels (up to 0.01 hPa). The ERA5 product extends from 1950 to the present. In this study the time range used was

the same as for the CMIP6 data, from 1985 to 2014. Valkonen et al. (2021) have shown that the ERA5 product performs well compared to the older reanalysis products."

- Reference typo on L134: (Wernli and Schwierz, 2006).

Corrected the typo

- Figure 1 caption should say "the Arctic"

Corrected

- Typo on L147: "recorded" instead of "recorder"

Corrected

- Abbreviation "ACE" is mentioned on L151 before being explained on L157. Please change

Corrected

- Extra ";" on L177.

Corrected

- Please rephrase the panel titles, e.g., in Fig. 3. This is true for almost every figure in this manuscript when the titles are given directly from the variable names in the models (it looks like it).

Corrected.

- On L196 I assume the given cyclone frequencies are *average* frequencies over the time period considered? Where are the trends in cyclone frequencies per season shown in the Figures/Tables (discussed on L200 onward)? If the authors decide to include the linear trends as lines or values, I would suggest adding them in all of the wide-basin figures.

Yes, these frequencies are based on table S1 and figure 2. The trends are shown in the trend matrix, figure 10a. We have added references to those figures and tables to make it more clear.

- "Figure" missing in the brackets at L238.

Corrected typo.

- Remove the second "is" in L290.

Corrected typo

- Number "6" on L431 a bit odd. I assume the authors refer to the Figure 6?

Corrected typo.

**Citation**: https://doi.org/10.5194/wcd-2023-2-RC2

Added citations:
*Lukovich et al., 2021:* https://doi.org/10.1175/JCLI-D-19-0925.1
*Schreiber and Serreze, 2020:* doi:10.1017/aog.2020.23
*Webster et al., 2018:* https://doi.org/10.1038/s41467-019-13299-8
*Boisvert et al., 2016:* https://doi.org/10.1175/MWR-D-16-0234.1
*Blanchard-Wrigglesworth et al., 2022:* https://doi.org/10.1029/2022JD037161
*Finnochio et al. (2022):* https://doi.org/10.1029/2020GL088338
*Koyama et al., 2017:* https://doi.org/10.1175/JCLI-D-16-0542.1
*Messori et al., 2018:* https://doi.org/10.1175/JCLI-D-17-0386.1
Parker et al. 2023: https://doi.org/10.1038/s41467-022-34126-7
*Holland and Stroeve, 2011:* Changing seasonal sea ice predictor relationships in a changing Arctic climate. *Geophys. Res. Lett.* **38**, L18501 (2011).
*Mortin et al., 2016:* https://doi.org/10.1002/2016GL069330